# TimeTalk uses single-cell RNA-seq datasets to decipher cell-cell communication during early embryo development

Longteng Wang [1,2], Yang Zheng[3], Yu Sun[3], Shulin Mao[2,4], Hao Li [3], Xiaochen Bo [2], Cheng Li [2✉] & Hebing Chen [3✉]

Early embryonic development is a dynamic process that relies on proper cell-cell communication to form a correctly patterned embryo. Early embryo development-related ligand-receptor pairs (eLRs) have been shown to guide cell fate decisions and morphogenesis. However, the scope of eLRs and their influence on early embryo development remain elusive. Here, we developed a computational framework named TimeTalk from integrated public time-course mouse scRNA-seq datasets to decipher the secret of eLRs. Extensive validations and analyses were performed to ensure the involvement of identified eLRs in early embryo development. Process analysis identified that eLRs could be divided into six temporal windows corresponding to sequential events in the early embryo development process. With the interpolation strategy, TimeTalk is powerful in revealing paracrine settings and studying cell-cell communication during early embryo development. Furthermore, by using TimeTalk in the blastocyst and blastoid models, we found that the blastoid models share the core communication pathways with the epiblast and primitive endoderm lineages in the blastocysts. This result suggests that TimeTalk has transferability to other bio-dynamic processes. We also curated eLRs recognized by TimeTalk, which may provide valuable clues for understanding early embryo development and relevant disorders.

[1] Peking University-Tsinghua University-National Institute of Biological Sciences Joint Graduate Program, School of Life Sciences, Peking University, Beijing 100871, China. [2] Center for Bioinformatics, School of Life Sciences, Center for Statistical Science, Peking-Tsinghua Center for Life Sciences, Peking University, Beijing 100871, China. [3] Institute of Health Service and Transfusion Medicine, Beijing 100850, China. [4] Yuanpei College, Peking University, Beijing 100871, China. ✉email: cheng_li@pku.edu.cn; chenhb@bmi.ac.cn

Early embryo development is a multi-level regulatory process in which fertilized egg undergoes rounds of cleavage to form a self-organized, hollow sphere structure called blastocyst[1]. At the beginning of the early embryo development, the paternal gametes sperm and maternal gametes oocytes fused to form a one-cell embryo called the zygote in fertilization. Then, the one-cell embryo zygote experiences rounds of cell cleavage division with maternal factor decay and zygotic genome activation (ZGA), which is called maternal to zygotic transition[2]. After the fourth cleavage division, the embryo begins to compact into blastomeres. With blastomere formation, there are two critical differentiation events: the first cell fate decision and the second cell fate decision. In the first cell fate decision from the 8-cell stage to the 32-cell stage, the cells in the embryo segregate into two lineages: tro-phectoderm (TE) and inner cell mass (ICM), while TE lineage will develop into the placenta. However, in the second cell fate decisions from the early-blastocyst stage to the late-blastocyst stage, the cells in the ICM lineage differentiated into epiblast (EPI) and primitive endoderm (PE) lineage. The EPI lineage will develop into the fetus, and the PE lineage will develop into the Yolk sac[1,3,4]. The phenomena intrigue evolving research on the mechanisms of the two fate decision events[4,5].

Cell-cell communication is a fundamental process in which, for a given cell, the membrane protein (i.e., receptor) is bound by the protein secreted by other cells or itself, called a ligand, inducing intracellular signaling responses[6]. Recently, many studies illustrated that cell-cell communication participates in the two fate-decision events during early embryo development. For example, during the first fate decision, LIF-JAK/STAT pathway-related pair *Lif-Lifr* is essential for ICM lineage maintenance[7]. Nevertheless, during the second fate decision, the communication between cells mediated by *Fgf4-Fgfr1* and *Fgf4-Fgfr2* interplay with master regulator Nanog and Gata6 promote PE lineage formation and maturation[1]. The progenitor cells in the ICM with high expression of *Gata6* express high levels of the FGF receptor genes *Fgfr1* and *Fgfr2*. Similarly, the progenitor cells in the ICM with high expression of *Nanog* express high levels of *Fgf4*. The binding of FGF4 secreted by Nanog^high cells to FGFR1 or FGFR2 in Gata6^high cells constructs a positive feedback loop with GATA6 to activate gene expression of primitive endoderm program[1,8]. Moreover, the cell-cell communication mediated by *Fgf4-Fgfr1* of Nanog^high cells promotes EPI exiting from pluripotency[9]. Thus, *Fgf4-Fgfr1* and *Fgf4-Fgfr2* regulate PE and EPI formation.

In conclusion, previous research indicates that cell-cell communication via ligand-receptor pairs has critical functions during early embryo development. We termed these early embryo development-related ligand-receptor pairs as eLRs. Studying eLRs will help us better understand cell-cell communication during early embryo development. However, the scope of eLRs and their influence on early embryo development is still lacking.

With the advancement of functional genomics research, more cell-cell communication *priori* knowledge resources, including ligand-receptor databases, have been accumulated in recent years[6]. On the other hand, the development of single-cell RNA sequencing (scRNA-seq) technology has made it possible to infer the cell-cell communication events between different cells. Consequently, integrating single-cell transcriptomic sequencing data and prior knowledge to infer cell-cell communication has emerged as a new research direction in bioinformatics. As for early embryo development studies, research performed scRNA-seq over each development stage to obtain the time-course scRNA-seq data to measure the dynamic changes in cell states and types. However, it is critical to note that definitive cell types do not emerge during early embryo development until blastocyst formation. However, commonly used tools for cell-cell communication inference, such as CellPhoneDB[10] and CellChat[11], are

designed to study cell-cell communication between given cell types and cannot meet the requirements of early embryo development research. Moreover, the need for multiple time points of cell-cell communication analysis during early embryo development and the need to elucidate potential causal relationships between cell-cell communication and gene regulatory networks during dynamic changes of early embryo development also pose challenges to existing cell-cell communication research.

To address these issues, we developed and applied a computational framework named TimeTalk for utilizing temporal series information to study the dynamics of autocrine signaling within the embryos and to identify early embryo development-related ligand-receptor pairs (eLRs) from integrated public time-course mouse scRNA-seq datasets. Our analysis identified 430 eLRs, including previously reported *Fgf4-Fgfr1* and *Fgf4-Fgfr2*. Additionally, we conducted thorough in silico analyses to test the involvement of identified eLRs in early embryo development. After validation, we found that the identified eLRs can be divided into six temporal windows. Furthermore, the GO analysis reveals that different temporal windows correspond with sequential early embryo development events. Moreover, we used Granger causality and network analysis to discover and check the potential regulation relationship between eLR and corresponding temporal TFs (tTFs). To broaden the application of TimeTalk to paracrine studies and other development processes, we further improved the framework, used it to investigate cell-cell communication in blastocysts, and reconstructed in-vitro embryo models named blastoids. The results indicated that the communication between EPI and PE lineages in blastoids involves shared core LR pairs and signaling pathways as in naturally developed blastocysts. In summary, the identified eLRs would be a valuable resource for better understanding early embryo development, and some would be targets to perturb some key early embryo development processes. In addition, we believed TimeTalk would be a helpful toolkit to study cell-cell communication in other development processes. TimeTalk is an R package that is available at https://github.com/ChengLiLab/TimeTalk.

## Results

**Curation of early-embryo development single-cell RNA-seq data sets for studying cell-cell communication.** To identify and study eLRs, we collected public early embryo development scRNA-seq datasets from the mouse MII-oocyte stage to the late blastocyst stage to ensure that scRNA-seq datasets represented every stage of early embryo development. In addition, to validate the quality of scRNA-seq data, we also collected public low-input RNA-seq data (Fig. 1a).

We first validated the quality of the scRNA-seq data. Each stage's overall gene expression distributions were similar for the low-input RNA-seq data and the pseudo-bulk single-cell RNA-seq data (Supplementary Fig. 1a–d; for low-input RNA-seq, gene expression was quantified by RPKM; for pseudo-bulk single-cell RNA-seq data, gene expression was quantified based on the average gene expression of cells in different stages). The principal component analysis (PCA) showed that the single-cell RNA-seq and low-input RNA-seq datasets were clustered together at each stage (Supplementary Fig. 1e). By comparing the expression profile of pseudo-bulk RNA-seq data derived from scRNA-seq data and low-input RNA-seq data, we found that the pseudo-bulk and low-input RNA-seq expression profiles of the samples from each stage were highly correlated. For example, the Pearson correlation coefficient of the bulk late 2-cell expression profile with the pseudo-bulk late 2-cell expression profile was 0.91 (Fig. 1b and Supplementary Fig. 2). Consider MII-oocyte (E0) scRNA-seq data (GEO accession: GSE38495)[12] and E1-E4.0

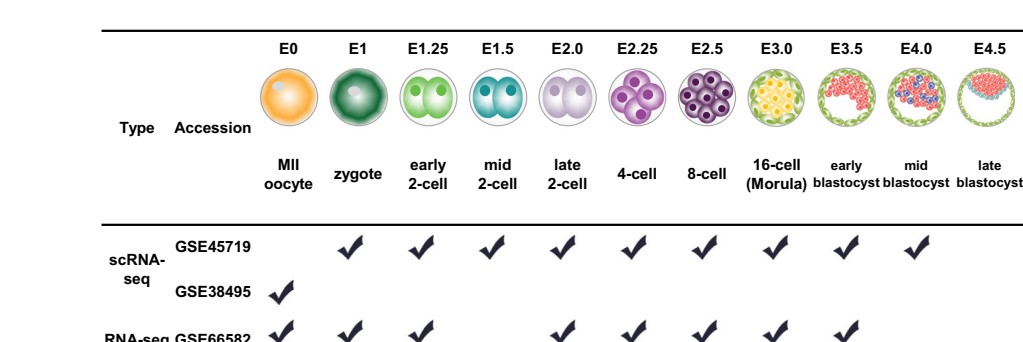

**Fig. 1 Curation of early-embryo development single-cell RNA-seq data sets for studying cell-cell communication. a** Metadata of the collected early embryo development datasets. **b** Heatmap of the Pearson correlation coefficients between bulk and pseudo-bulk data. **c** tSNE plot of the integrated single-cell RNA-seq data. **d** Feature plot of the different marker genes. The unit for color gradient keys is LogNormalize, which involves dividing the feature counts for each cell by the total counts for that cell and then multiplying it by the scale.factor. This value is then transformed using the natural-log function and adding 1. **e** Pseudotime ordering of collected early embryo cells reveals development trajectory. The top of the figure was colored by development stages, and the bottom was colored by pseudotimes.

scRNA-seq datasets (GEO accession, GSE45719)[13], which were obtained from two different studies. To ensure that potential batch effects could be disregarded, we performed the kBET analysis[14]. The results of the kBET analysis suggested that the batch effects were negligible (see Supplementary Fig. 3). Therefore, we can merge the two datasets without performing batch correction. Next, we put the scRNA-seq data into the Seurat toolkit to perform t-SNE dimension reductions (Fig. 1c) and check each stage's marker gene expression (Fig. 1d). Upon checking the marker gene expression, and the *monocle2* package[15] was utilized to perform trajectory analysis and arrange the cells along the constructed trajectory (Fig. 1e).

In summary, the curated early embryo scRNA-seq datasets recapitulate the early embryo development process.

**TimeTalk is a computational framework that utilizes temporal series information for eLR identification**. After building the comprehensive transcriptomics datasets, we began to identify eLRs. First, we obtained the ligand-receptor list from the comprehensive CellTalkDB database[16]. Then, we studied the gene expression distribution of ligands and receptors across developmental stages. We found that the expression levels of ligands and receptor genes were lower (Supplementary Fig. 4a) and more dynamic (Supplementary Fig. 4b, c) in comparison with the overall trends of gene expression at each stage. Furthermore, the previously reported eLRs exhibited significant co-variation changes along the pseudotime (Supplementary Fig. 5).

According to previous studies, well-studied eLR pairs *Fgf4-Fgfr1* and *Fgf4-Fgfr2* interplay with transcription factors *Sox2*, *Nanog*, *Oct4 (Pou5f1)*, *Gata6* to form a complex regulatory network to guide the ICM to differentiate into EPI or PE lineage[1] (Fig. 2a). Therefore, after ranking the cells according to pseudotime inference by the *monocle2* package, we investigate the expression of *Fgf4*, *Fgfr1, Fgfr2*, and related transcription factors *Gata6*, *Pou5f1*, Sox2, and *Nanog* (Supplementary Fig. 6). We found that the gene expression of *Fgf4* and *Fgfr2* were negatively correlated (Fig. 2b). Moreover, we calculate the geometry means of *Fgf4* and *Fgfr2* gene expression (Interaction Score, or can be noted as IS)[17] to quantify the cell-cell communication activity of *Fgf4-Fgfr2*. We found that the IS of *Fgf4-Fgfr2* was correlated with *Gata6* gene expression (Fig. 2c).

These results indicated that eLR gene pairs show coordinated expression patterns with core transcription factors during early embryo development. According to these observations, we developed an early-embryo ligand-receptor screening strategy, TimeTalk, to identify eLRs with the following steps. (1) Building trajectory from integrated single-cell RNA-seq datasets, ordering the single-cells along the trajectory. (2) A ligand-receptor database was used to identify the co-varying LR pairs from the pseudotime time series data (Supplementary Fig. 7, details are described in the Methods section). Meanwhile, the transcriptional regulatory network is reconstructed from scRNA-seq data to identify the temporal TFs (tTFs) during early embryo development. (3) Cluster the tTFs. (4) Inference of the regulatory relationship between the different clusters of tTF and co-varying LR pairs through the Granger causality approach by Granger causal test. (5) The LR pairs that passed the Granger causal test were considered as forward (LR regulates TF), backward (TF regulates LR), and feedback (LR and TF were mutually regulated) (Fig. 2d, e, details are described in the Methods section).

According to TimeTalk workflow, we identified 430 eLRs, and as a by-product of eLR identification, we identified 229 tTFs during early embryo development. The 229 tTFs were clustered into six distinct groups (Supplementary Data 1, Supplementary Fig. 8). Subsequent analysis revealed that the previously reported

eLR Bmp4-Bmpr2 also exhibited varying Granger causality relationships with different clusters of tTFs (Supplementary Fig. 9). Cluster C1 tTFs consist of maternal factors such as *Nfya*[18], *Obox5*, and *Atf2* and they are enriched in GO terms like "oogenesis" and "germ cell development". Cluster C2 contains ZGA genes like *Zscan4f* and is enriched with GO terms like "histone methylation" and "protein methylation". Cluster C3 comprises TFs that play a role in establishing heterogeneity. For example, *Sox21* contained in cluster C3 was reported to be involved in fate biases starting from the 4-cell stage[19]. Cluster C4 is composed of TFs like *Ctcf* involved in chromatin reorganization and enriched in GO terms like "histone methylation", "chromatin silencing", and "chromatin assembly or disassembly", "DNA conformation change". Cluster C5 is composed of tTFs involved in blastocyst development like *Gata6*, *Tead4*, *Cdx2*, *Nanog*, and *Pou5f1* and enriched in GO terms like "trophectodermal cell differentiation", "blastocyst development", "blastocyst formation", "blastocyst growth". Cluster C6 is also composed of tTFs involved in blastocyst development like *Sox2* and enriched GO term "embryonic organ development", and "regulation of epithelial cell differentiation" (Fig. 2f, g, Supplementary Data 1). Additionally, we found that C5 tTFs regulate *Fgf4-Fgfr2* activity (Fig. 2h, i, Supplementary Fig. 10).

In summary, the tTFs identified here provided potential targets for manipulating early embryo development at different temporal windows.

**The predicted eLRs can be verified from extensive in-silico validations**. Next, we validated the eLRs identified by TimeTalk from multiple in-silico strategies.

**The candidate eLRs predicted by the TimeTalk workflow contain reported eLRs**. We ranked the identified eLRs pairs according to their PCC values. A literature review yielded evidence that the top 30 positively correlated eLRs and the top 30 negatively correlated eLRs participate in early embryo development (Fig. 3a, Supplementary Data 1). The eLRs identified using TimeTalk also included ligands and receptors with established roles in early embryo development. For example, the candidate eLRs contain *Bmp6-Bmpr2*, which has been reported to regulate extra-embryonic lineage development[20].

**The enrichment of essential gene in the candidate eLRs indicate that the genes composed of predicted eLRs were mainly required for early embryo development**. Essential genes are required for organisms to survive and fit in with the environment[21,22]. A loss of function of any essential gene generally produces serious deleterious effects, including embryonic lethality[23]. Thus, essential genes among the eLRs identified using TimeTalk are likely required for early embryo development. We obtained the mouse essential gene list provided by the DEG 10 database[24] and computed essential gene set enrichment ratios for all protein-coding genes (all), ligand and receptor genes (LRs), and eLRs. In comparison with all protein-coding genes, the set of essential genes was more significantly enriched with LRs and eLRs (Fig. 3b, Supplementary Data 1, "LR" vs. "all", $p = 1.628 \times 10^{-111}$; "eLR" vs. "all" $p = 3.049 \times 10^{-85}$, $p$-values were obtained by hypergeometric test, upper tail). This observation was consistent with the principle that the proteins encoded by ligand and receptor genes required cells to respond to external stimuli. In addition, the set of essential genes was more enriched in eLRs in comparison with LRs (Fig. 3b, Supplementary Fig. 11a, Supplementary Data 1, "eLR" vs. "LR", $p = 4.344 \times 10^{-14}$, $p$-values were obtained by hypergeometric test, upper tail). The enrichment of essential gene in the candidate eLRs indicate the

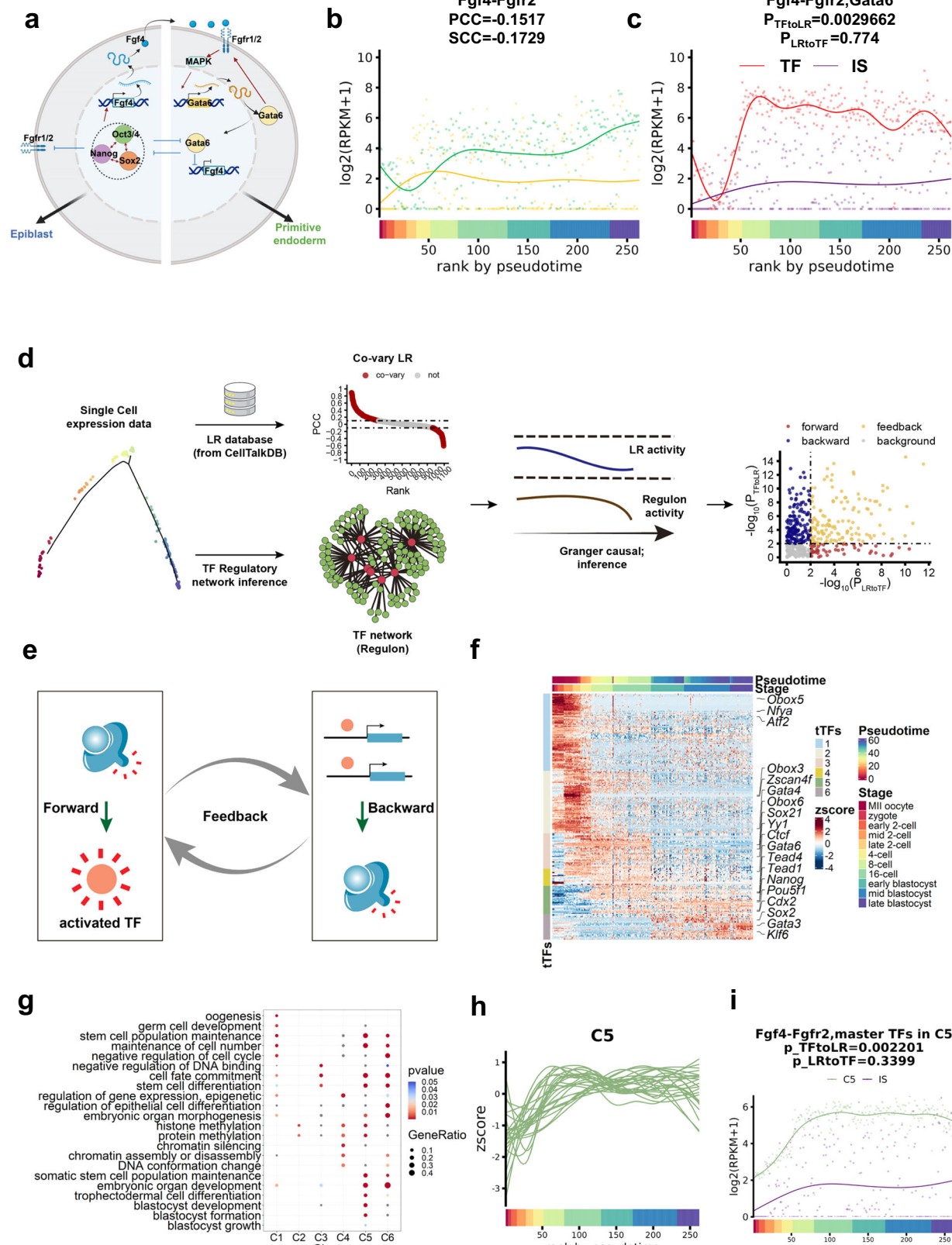

genes composed of predicted eLRs were mainly required for the organism's fitness.

**The absence of housekeeping genes in the candidate eLRs suggests that the genes comprising the predicted eLRs were not required for individual cell survival.** Housekeeping genes are stably expressed genes that maintain basic cellular functions regardless of cell type or developmental stage[25]. In contrast, the genes comprising eLRs should be dynamically expressed to coordinate early embryo development. This result led us to suspect that the identified eLR genes had been absent of housekeeping genes. Therefore, we obtained the housekeeping gene list

**Fig. 2 TimeTalk is a computational framework that utilizes temporal series information for eLR identification. a** The Fgf4-Fgfr1 and Fgf4-Fgfr2 work together to regulate a core pluripotency gene network. Red lines with arrows show positive regulation, while blue lines with inhibitory marks indicate negative regulation. The black lines represent the gene expression process. This figure was created using BioRender (https://app.biorender.com/). **b** The ligand and receptor genes in *Fgf4-Fgfr2* co-vary during development. PCC, Pearson's correlation coefficient, SCC, Spearman's correlation coefficient. **c** The dynamics of interaction score of *Fgf4-Fgfr2* and transcription factor Gata6. The *p*-value of p_TFtoLR and p_LRtoTF were calculated by the Granger causal test, revealing the potential causal relationship between TF and LR. **d** The workflow of the preliminary version of TimeTalk. **e** Illustration of the relationship between the activated LR and TFs. Forward means LR activates TF, backward means TF regulates LR, and feedback means LR and TF mutually regulate each other. **f** The 229 temporal TFs (tTFs) revealed by master regulator analysis were divided into six clusters. The right side marked representative TFs of each group. **g** The GO analysis of six clusters of tTFs. **h** The dynamics of tTFs in C5 gradually decreased during development. **i** The line plot depicts the dynamic variations in the interaction score of *Fgf4-Fgfr2* and the activity of C5 tTFs.

provided by the HRT Atlas v1.0 database[26] and calculated the housekeeping gene enrichment ratio for all protein-coding genes (all), ligand and receptor genes (LRs), and eLR genes (eLRs). As expected, in comparison with the set of all protein-coding genes, the set of eLR genes was more depleted of housekeeping genes (Fig. 3c, Supplementary Data 1, "LR" vs. "all", $p = 1.944 \times 10^{-47}$; "eLR" vs. "all", $p = 5.986 \times 10^{-16}$, *p*-values were obtained by hypergeometric test, lower tail). Moreover, the housekeeping gene enrichment ratio of the set of all ligand and receptor genes was not significantly different from that of the identified eLRs (Fig. 3c, Supplementary Fig. 11b, Supplementary Data 1). Together with the essential gene enrichment results, these findings suggest that the eLR genes identified using TimeTalk were more important for the organism's fitness, rather than for the survival of an individual cell.

**Many genes composed of the candidate eLRs were activated by the ZGA process, and the cell-cell communication mediated by the candidate eLRs required the ZGA process.** ZGA is the embryogenesis process in which maternally provided factors are replaced by factors supplied by zygotic transcription[27]. Transcription of the ZGA gene is required for lineage specification and provides substrates for the initiation of gastrulation[27,28]. Therefore, ZGA is vital for early embryo development. Thus, we hypothesized that ZGA genes were more enriched in the identified eLR genes than the LR genes. The ZGA gene list was obtained from ref. [18], and the ZGA gene enrichment ratio was calculated for the protein-coding genes (all), ligand and receptor genes (LRs), and eLR genes (eLRs). The enrichment ratio of the ZGA gene in the protein-coding gene set was not significantly different from that of the LR gene set (Fig. 3d, Supplementary Data 1, Supplementary Fig. 11c, $p = 0.09293$, *p*-values were obtained by hypergeometric test, upper tail). However, the ZGA gene enrichment ratio of the eLRs was significantly different from that of the protein-coding gene set (Fig. 3d, Supplementary Fig. 11c, Supplementary Data 1, $p = 0.001159$, p-values were obtained by hypergeometric test, upper tail). In addition, the ZGA gene enrichment ratio of the eLRs was higher than that of the LR gene (Fig. 3d, Supplementary Fig. 11c, Supplementary Data 1, $p = 0.00279$, *p*-values were obtained by hypergeometric test, upper tail). These findings indicate that the set of eLRs identified using TimeTalk was enriched with ZGA genes when compared to both LR genes and protein-coding genes. This result suggests that the many genes composed of candidate eLRs were activated during the ZGA process.

It has been established that ZGA is essential for preimplantation in mice[29]. Considering that eLR-mediated cell-cell communication is essential for early embryo development, we hypothesized that the signaling mediated by specific eLR pairs might require ZGA. Accordingly, we obtained ZGA inhibition RNA-seq datasets[30] and calculated interaction scores[17] (the calculation of interaction score is described in the Methods section) to quantify the cell-cell communication strength

mediated by eLR pairs. As a result, ZGA inhibition altered the cell-cell communication strength of 56 eLR pairs (Fig. 3e, Supplementary Fig. 12). Thus, the establishment of candidate eLRs mediated cell-cell communication correlated with the ZGA process. Furthermore, compared to non-eLR pairs, eLR pairs show less impact from ZGA inhibition (Supplementary Fig. 13, Supplementary Data 1). This suggests that eLR pairs possess additional regulation mechanisms beyond ZGA.

In summary, the results above demonstrate that the eLR pairs identified by TimeTalk have several essential characteristics. First, many highly ranked eLR pairs have been reported in the literature in relevant contexts. Second, the identified eLR pairs are enriched with the essential gene and the ZGA gene but depleted of the housekeeping gene. Third, ZGA inhibition disrupted some types of cell-cell communication mediated by the identified eLR pairs. Finally, these characteristics supported the involvement of the identified eLR pairs in early embryo development.

**The interplay between the tTFs and eLR orchestrates early embryo development.** As it is previously reported, the interplay between the eLR *Fgf4-Fgfr2* and tTF *Gata6* formed a positive regulatory network to guide the separation of PE and EPI lineage during early embryo development[8]. However, no comprehensive profiling of the relationship between eLR and tTF has been conceived. Therefore, we hypothesized that investigating the interplay between eLR and TF will help us to understand the gene regulatory network to control mouse embryo development.

The identified eLRs can be clustered into 6 clusters associated with different groups of temporal TFs. Cluster 1 eLRs have a backward relationship with C1 tTFs. This result implies that maternal tTFs control the cluster 1 eLRs. This conclusion is supported by the observation that cluster 1 eLRs exhibit a higher ratio of the maternal gene (Supplementary Fig. 14a). Cluster 3 eLRs exhibit a higher ratio of ZGA genes (Supplementary Fig. 14b). In addition, cluster 1 eLRs have a feedback relationship with C5 tTFs. Interestingly, cluster 4 eLRs have a forward relationship with C1-C6 tTFs. Cluster 5 eLRs have feed relationships with C1, C2, C3, and C4 and backward relationships with C6. Cluster 6 eLRs have a backward relationship with C6 tTFs (Fig. 4a, Supplementary Data 2). Clusters 1, 2, 3, and 4 enriched MAPK signaling pathways (Fig. 4b, Supplementary Data 2). This pathway is involved in cell differentiation during early embryo development[31,32].

Cluster 4,5,6 eLRs enriched KEGG term "Signaling pathways regulating pluripotency of stem cells". Besides, clusters 4 and 6 enriched the "Hippo signaling pathway" related to trophectoderm lineage formation[33] (Fig. 4b, Supplementary Data 2). Upon analyzing the enrichment results, we discovered that cluster 4 eLR was enriched in the Hippo signaling pathway with 23 eLRs consisting of genes such as *Cdh1, Wnt3a, Areg, Gdf5, Wnt5a, Wnt7a, Fgf1, Tgfbr1, Fzd2, Itgb2, Fzd5, Fzd1, Fzd4, Fzd9, Bmpr1b, Bmpr2,* and *Fzd3*. Similarly, cluster 6 eLR was found to be enriched in the Hippo signaling pathway with six eLRs consisting

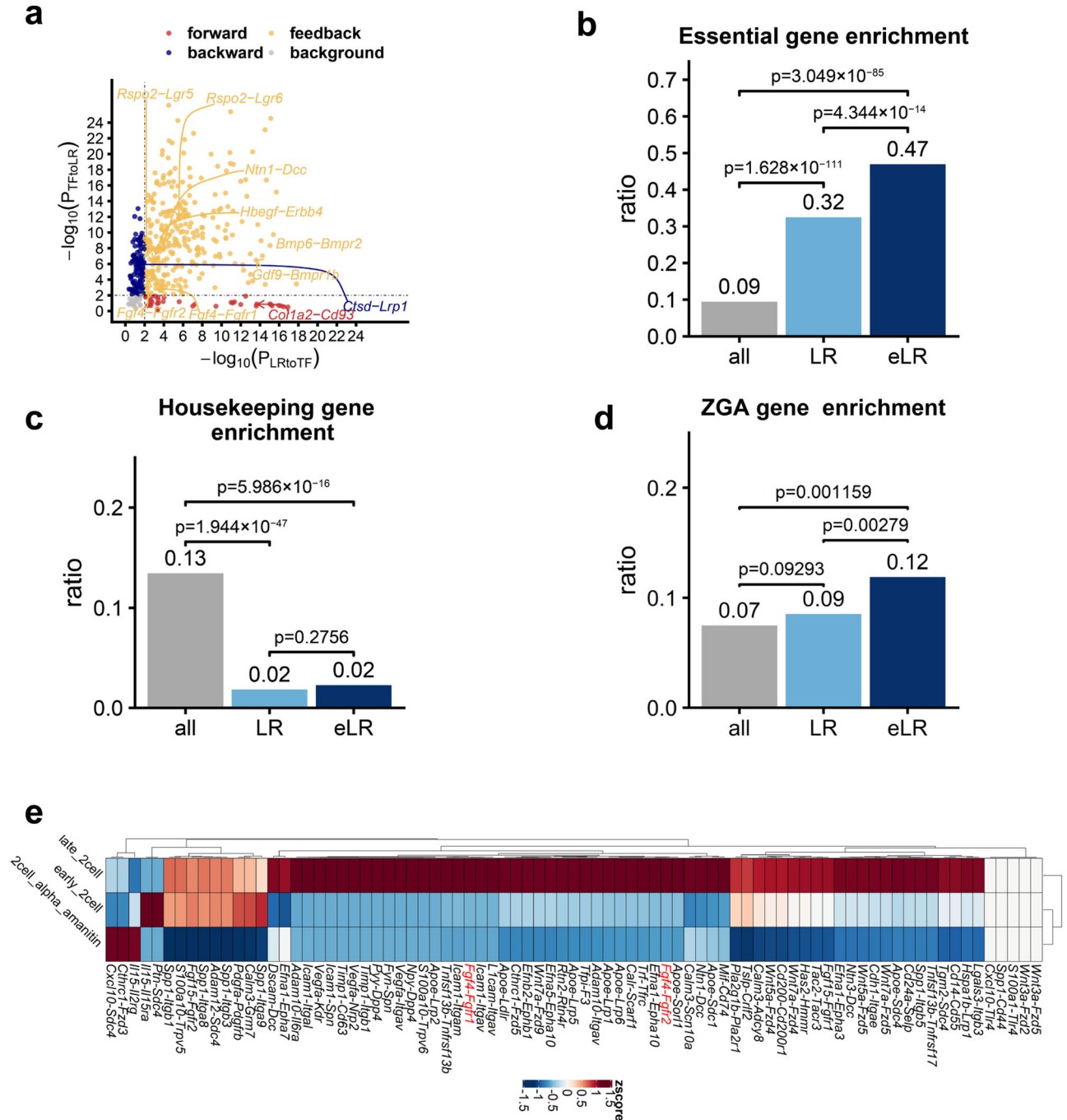

**Fig. 3 The predicted eLRs can be validated through extensive in-silico analyses. a** The predicted eLRs; each dot represents an eLR. Previously reported eLR pairs were highlighted. **b** The enrichment ratio of essential genes in the sets of all protein-coding genes (all), ligands and receptors (LRs), and eLRs. **c** The enrichment ratio of housekeeping genes in all protein-coding genes (all), ligands and receptors (LRs), and eLRs. **d** The enrichment ratio of ZGA genes in the sets of all protein-coding genes (all), ligands and receptors (LRs), and eLRs. We used the hypergeometric test to assess the results' significance in (**b–d**). **e** Heatmap of the interaction scores of some eLRs that were down-regulated by the ZGA inhibition, the ZGA-inhibited RNA-seq data downloaded from GSE71434.

of genes like *Wnt3a, Wnt7b, Fzd7, Fzd5, Fzd3*, and *Fzd6* (Supplementary Fig. 15a). Most of these genes are known to be involved in the TGF-β and Wnt signaling pathways, which have crosstalk with Hippo signaling pathway[34]. The KEGG database also highlights this crosstalk between the Hippo, Wnt, and TGF-β signaling pathways (see web link from the KEGG database: https://www.genome.jp/pathway/mmu04390). An investigation was conducted to determine whether there is a connection between eLR activity and the Hippo signaling pathway. It was

discovered that the C5 tTF includes Tead2 and Tead4, which are part of the core Hippo pathway gene sets[35]. Therefore, *Tead2* and *Tead4* can be indicators of Hippo signaling pathway activity. As for *Tead2*, the eLR *Wnt3a-Fzd5* exhibited an increase in activity over pseudotime, with the Granger test indicating that *Tead2* was responsible for activating *Wnt3a-Fzd5* activity (Supplementary Fig. 15b). Prior research has demonstrated that both the Wnt3a and Fzd5 genes are crucial for subsequent trophectoderm lineage development[36]. Our findings, combined with this previous

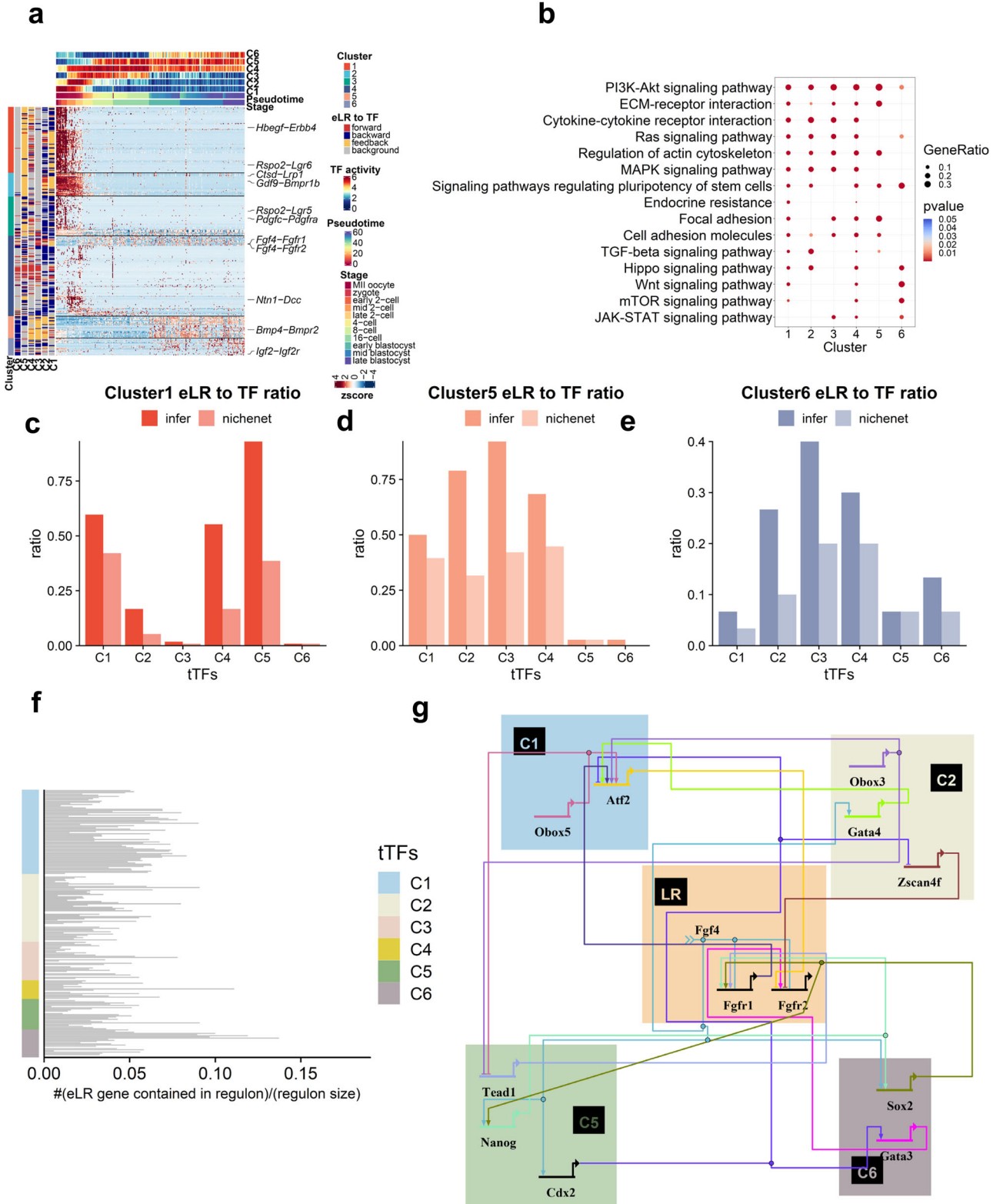

**Fig. 4 The interplay between the temporal transcription factor and eLR orchestrates early embryo development. a** Heatmap of the interaction scores of all identified 430 eLRs can be grouped into 6 clusters. In the row annotation of the heatmap, the "cluster" column illustrates the classification of the eLR, C1, C2, C3, C4, C5, and C6 columns illustrate the relationship of each eLR to the C1, C2, C3, C4, C5, C6 regulons. In the column annotation of the heatmap, the "Stage" row illustrates the development stage of every single cell, the "Pseudotime" row illustrates the pseudotime, C1, C2, C3, C4, C5, C6 row illustrates the activity of every single cell. **b** The KEGG analysis of eLRs clustered for each clusters. **c–e** The cluster 1, cluster 5, and cluster 6 eLR to TF ratio, the signaling network provided by NicheNet can verify the ratio. **f** The distribution of eLR genes across the regulons of various tTFs. Each row in the figure corresponds to a specific tTF, with the horizontal bar indicating the ratio of eLR genes present in its regulon. **g** Fgf4-Fgfr1 and Fgf4-Fgfr2 interacted with different tTFs, forming an organized regulatory network.

research, demonstrate that Hippo signaling pathway activation leads to eLR activation, indicating a potential role for the Hippo signaling pathway in regulating trophectoderm development by activating eLR activity. The eLR *Cthrc1-Fzd3* exhibited a peak expression between the 4-cell stage and 16-cell stages with respect to *Tead4* expression, suggesting that *Tead4* has a feedback relationship with *Cthrc1-Fzd3* activity (Supplementary Fig. 15c). Previous studies indicate that *Cthrc1* selectively activates the planar cell polarity pathway of Wnt signaling by stabilizing the Wnt-receptor complex[37]. Additionally, *Cthrc1* promotes trophoblast growth, migration, and invasion through reciprocal Wnt/β-catenin regulation[38]. Our findings suggest that the cell-cell communication triggered by *Cthrc1* may also interact with the Hippo signaling pathway to promote trophectoderm lineage formation.

In addition, we verified regulation from eLR to tTF by the signaling network created from NicheNet. (Fig. 4c–e, Supplementary Fig. 16a–c, Supplementary Data 2). Besides, we investigate regulation relationships from tTFs to eLRs and find that C1 and C6 regulate more eLRs than the other cluster of tTFs (Fig. 4f, Supplementary Fig. 17, Supplementary Data 2). We used BioTapestry[39] to illustrate well-studied eLRs Fgf4-Fgfr1, Fgf4-Fgfr2 interplay with temporal TFs like Sox2 and Nanog form a complex regulatory network (Fig. 4g).

In conclusion, the interplay between the tTFs and eLR orchestrates early embryo development.

**TimeTalk workflow can be updated to decipher cell-cell communication in blastoid.** Blastoids are a valuable system for studying early embryo development in vitro[40]. Therefore, we hypothesized it would be helpful to study the cell-cell communication between different blastoid lineages to check blastoids' fidelity in modeling naturally developed blastocysts.

However, some obstacles exist to employing TimeTalk for investigating cell-cell communication in blastoids. The presence of different cell lineages in blastoids and blastocysts requires the investigation of paracrine signaling between distinct cell types. Moreover, the available public data on blastoid research primarily comprise droplet-based single-cell RNA-seq datasets, which can be relatively sparse. This sparsity poses challenges when building transcriptional networks from the available data. These obstacles may make it difficult to apply the TimeTalk workflow directly to the study of blastoid cell-cell communication. Therefore, we update the TimeTalk workflow with the following procedures: (1) As the *monocle3*[41] package can reconstruct complex development trajectories from scRNA-seq datasets, we used monocle3 to replace *monocle2* to order cells in each cell type. (2) Next, to study the correlation of ligand and receptor expression between two cell types, we interpolated pseudotime the time series by *CellAlign*[42] to get equal sample points for two cell types. (3) We used Spearman's correlation coefficient to get co-varying LR pairs between two cell types. (4) We perform zero-preserved imputation by the ALRA algorithm[43]. (5) The impute matrix was used to build the transcriptional network and perform master regulator analysis by RTN package[44]. (6) Performing the Granger causal test to calculate the Granger causality between the gene expression of master TFs from the receiver cell types and interaction score of co-varying LR pairs from the interpolated pseudotime time series. (7) The LR pairs that passed the Granger causal test were considered forward (LR regulates TF), backward (TF regulates LR), and feedback (LR and TF were mutually regulated) (Fig. 5a).

Next, we re-analyzed a public scRNA-seq data set containing EPS-blastoids and natural blastocyst[45]. Consistent with the original study, the EPS-blastoid captures the main lineage EPI,

PE, and TE as blastocyst (Fig. 5b, Supplementary Fig. 18a–c). In addition, we found that the ICM-like lineage in EPS-blastoids also expressed 2-cell markers like *Zscan4c*, *Zscan4d*, and *Zscan4f* (Supplementary Fig. 18d, e). Therefore, we named these cells 2C-like cells. Besides, there are also some intermediate lineages (Supplementary Fig. 18f, g). As we annotated the cell types across different lineages, we used monocle3 to reconstruct trajectories and order cells in each lineage (Fig. 5c, Supplementary Data 3).

We are currently utilizing the updated version of TimeTalk to investigate cell-cell communication between the EPI-PE lineage in blastocysts and EPS-blastoids. As described in the previous paragraph, the interpolation and subsequent Granger causal test procedures in TimeTalk rely on three key parameters: window size of the interpolation (winsz), number of desired interpolated points (numPts), and the order of lags to include in the auxiliary regression (lag). We have set the default parameter values as winsz = 0.1, numPts = 200, and lags = 1, to identify 311 ligand-receptor pairs that mediate cell-cell communication between the epiblast and primitive endoderm in blastocyst cells. (Supplementary Data 3). TimeTalk is quite sensitive to the parameter winsz (Supplementary Fig. 19a), but relatively stable to numPts and lags parameters (Supplementary Fig. 19b, c). Moreover, it remains relatively robust when using the three critical parameters within certain ranges (Supplementary Fig. 19d). Through our analysis of TimeTalk results, we have discovered a feedback relationship between *Fgf4-Fgfr2* and *Gata6* with regards to cell-cell communication between the epiblast and primitive endoderm (p_LR_to_TF = 0.0184, p_TF_to_LR = $4.74 \times 10^{-5}$, p-values were calculated by granger causal test Supplementary Fig. 19e). This result is consistent with previous research[46], which demonstrated that *Nanog* in the epiblast upregulates the expression of *Fgf4*. The secreted *Fgf4* from epiblast induced *Fgf4-Fgfr2* interaction in primate endoderm cells to release Gata6 expression through ERK signaling. *Gata6* then potentiates the upregulation of *Fgfr2*, and the feedback loop reinforces a primitive endoderm fate. Furthermore, the interpolation result can suggest that *Nanog* upregulates *Fgf4* expression, but *Fgf4* cannot regulate *Nanog* expression in the epiblast (SCC = 0.640, p_L_to_TF = 0.657, p_TF_to_L = 0.025, p-values were calculated by granger causal test, SCC means Spearman correlation coefficient, Supplementary Fig. 19f). Additionally, the interpolation result could also suggest the feedback relationship between *Fgfr2* and *Gata6* (SCC = 0.274, p_R_to_TF = $1.15 \times 10^{-10}$, p_TF_to_LR = $9.19 \times 10^{-5}$, p-values were calculated by granger causal test, Supplementary Fig 19g).

The Venn diagram shows that blastoid and blastocyst share common LR pairs identified by TimeTalk (Fig. 5d, Supplementary Data 3). These shared LR pairs include well-studied pairs *Fgf4-Fgfr1* and *Fgf4-Fgfr2*. Moreover, the genes composed of LR pairs identified by TimeTalk enriched common signaling pathways involved in blastocyst formation. The notable examples are that both blastocysts and blastoids are enriched with "Signaling pathways regulating pluripotency of stem cells", "TGF-beta signaling pathway", "MAPK signaling pathway" and "Hippo signaling pathway" (Fig. 5e, Supplementary Data 3).

To the best of our knowledge, these results consistent with the previous results of blastoid's fidelity in modeling cell-cell communication in balstocyst[40], but from a different computation perspective.

**Discussion**
Early embryo development is a gradual and continuous process involving increasing cell heterogeneity. It is worth noting that definite cell types do not occur until blastocyst formation during this process. However, most computation toolkits for studying cell-cell communication, such as CellPhoneDB[10] and CellChat[11],

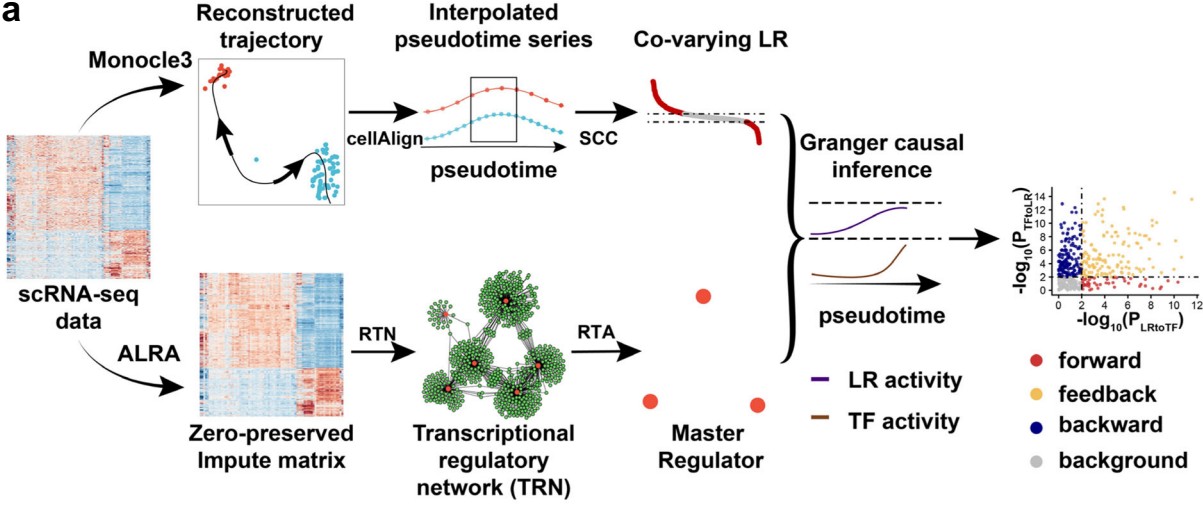

**Fig. 5 The extended TimeTalk workflow for deciphering cell-cell communication in blastoid. a** The updated version of TimeTalk. **b** The re-analysis of the EPS-blastoid dataset. Trajectories were plotted in the UMAP embeddings. The color bar on the right side indicates the pseudotime. **c** The percentage of different cell types in naturally developed blastocysts and EPS-blastoids. **d** The Venn diagram shows the overlap of LR pairs in blastocyst and blastoid. There, EPI lineage as a sender cell and PE lineage as a receiver cell. **e** The KEGG analysis of genes consists of LR pairs mediated EPI-PE communication.

were designed to investigate cell-cell communication in defined cell types, making it challenging to apply these tools to study early embryo development. To tackle these problems, it's essential to create a fresh computational workflow. We have developed the TimeTalk workflow, which utilizes temporal information to infer cell-cell communication during early embryo development. TimeTalk uses trajectory analysis and the Granger causal test to examine ligand-receptor pairs with causal relationships to master transcription factors during development, thereby elucidating the potential causal relationship between ligand-receptor interaction and TF activity. To study cell-cell communication in blastoids, we introduce an interpolated pseudotime time series strategy to perform the Granger causal test of two unequal length time series to infer cell-cell communication by different types of cells, making the TimeTalk workflow applicable to paracrine signaling in various development processes. In summary, TimeTalk is suitable for studying developmental processes, particularly when scRNA-seq datasets at different time points are available.

It should be admitted that the regulation relationship between ligand-receptor interaction and transcription factors inferred by the TimeTalk should be confirmed by further wet experiments. Given the absence of high temporal resolution data, we utilized two models based on theoretical Granger causality to create simulation datasets. These datasets were then employed to examine how the interpolation strategy might impact the conclusion of Granger causality. (Supplementary Note 1). According to our simulation results, the Granger causality conclusion is affected by the choice of parameter winsz (Supplementary Figs. 20 and 21). Thus, as presented in this manuscript, we acknowledge the need for additional methods to validate the inferred potential regulatory relationship between ligand-receptor interactions and transcription factors. Regrettably, the scarcity of high temporal resolution datasets hinders our ability to address this issue. Despite this limitation, we have acknowledged that our causal analysis serves as a helpful reference, and we intend to improve our approach in the future with more accurate data. Furthermore, we advise users in the TimeTalk documentation to conduct additional data analysis and experiment with various parameter combinations to achieve more dependable results when utilizing TimeTalk for cell-cell communication analysis.

It is noteworthy that TimeTalk can be employed to investigate normal developmental processes and explore cell-cell communication during disease progression. For example, in the study of cancer development and progression, the accumulation of longitudinal scRNA-seq data in cancer research[47,48] has made it possible to apply the TimeTalk framework to comprehend the cell-cell communication between cells within the tumor and its surrounding environment that promote cancer development and progression, ultimately aiding the development of effective cancer treatments.

It should be noted that the current version of the TimeTalk framework is designed for cell-cell communication mediated by ligand-receptor interactions. In contrast, other types of communication may not be accounted for. Extracellular vesicles (EVs) are an example of such communication. EVs, which are lipid bilayer structures that transport diverse biological cargo such as nucleic acids and proteins[49,50], are secreted from various mammalian cell types and hypothesized to mediate long-range cell-cell communication[51,52]. EVs are believed to have an influential role in both normal physiological and pathological processes[53], much like ligand-receptor interactions. However, previous EV profiling technologies have focused on the cargo of EVs rather than the molecules responsible for their secretion, making it difficult to decipher the function of EV-mediated cell-cell communication[54,55]. Recently, several studies attempted to develop single-cell profiling technology to detect EV secretion

and link it to disease progression[56,57]. These technological advancements can help extend the TimeTalk framework to encompass EV-mediated cell-cell communication and provide insights into how it participates in development and disease.

In this study, the identified eLRs would be a valuable resource for better understanding early embryo development. Furthermore, the further study of eLRs would bring new knowledge of how different cells coordinate each other's behavior to self-organize the embryo structure and give a new life. Moreover, the interplay between the eLRs and tTFs makes us block or enhance eLRs to manipulate tTFs activity or tune tTFs expression to regulate the downstream eLR activity to manipulate the cell-cell communication.

The process of co-evolution involves a heritable change in one entity that creates selective pressure for a change in another entity. Such entities can vary from nucleotides and amino acids to proteins, entire organisms, and potentially even ecosystems across the evolutionary time[58]. Co-evolution analysis was conducted on both eLR and non-eLR pairs, and the results indicate a similar distribution of co-evolution trends between the two groups (Supplementary Fig. 22). These results indicate that eLR and non-eLR are exposed to the same selection pressure.

This study assumed autocrine regulation of early embryo developmental stages during eLR screening. This hypothesis was reasonable for two reasons. First, before implantation, cells in the embryo are constrained by a glycoprotein shell called the zona pellucida[1,59]. Thus the embryo may be treated as a whole cell at each stage during pre-implantation. Second, previous studies illustrated that autocrine signaling is a substantial feature of the cell-to-cell communication network[60,61]. However, as crosstalk between the embryo and the maternal environment is essential for embryo development[62], this hypothesis precludes our study from exploring maternal-embryo interactions during early embryo development. Nevertheless, the rapid growth of spatial transcriptomics applications in the maternal microenvironment[60] and tissue engineering strategies[63] allowing researchers to mimic maternal-embryo interactions will facilitate experiments exploring how eLRs mediate maternal-embryo crosstalk during early embryo development.

Research in Drosophila brain development recently proved that different neuron cell types were successively generated by the sequentially expressed temporal transcription factors (tTFs)[64,65]. This study illustrates that the interplay of eLRs and tTFs can be divided into different temporal windows. Moreover, the successive activation of tTFs and subsequent correspondent activation of cell-cell communication by eLRs indicates a wave-like mechanism to guide embryo development from a zygote (Fig. 6).

Blastoid is an ex-vivo reconstruction of blastocyst-like structure from stem cell lines to model blastocyst development[40,66,67]. Generating a blastoid close to the naturally developed blastocyst is an excellent way to model early embryo development[68]. In this study, we illustrate the fidelity of blastoids in modeling blastocysts from the cell-cell communication perspective. In addition, it should be noted that blastoid can be used to validate the function of identified eLRs.

Overall, the recognized eLR list provides valuable clues for the community to understand early embryo development and developmental diseases. Furthermore, TimeTalk would be a helpful tool for studying cell-cell communication in other developmental processes.

## Methods

**Ethical approval statement**. After a thorough review of the sequencing studies utilized in the article, we can assure that the mouse experiments were carried out with appropriate ethical approval.

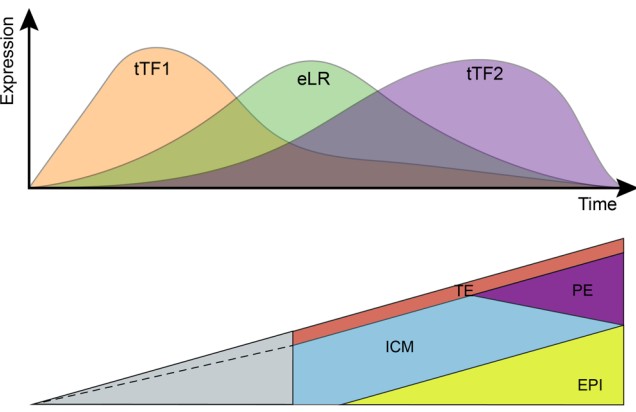

**Fig. 6 The temporal expression of tTFs and eLR act as molecular switches to guide early embryo development.** In the top panel, the first wave expression of temporal TFs (tTF1) triggers the expression eLR then the activity of eLR triggers the second wave of tTFs expression (tTF2). The bottom panel, the dynamics of tTF and eLR guide the early embryo development.

## Early embryo scRNA-seq preprocessing and quality control.

Publicly available pre-implantation (oocyte, zygote, early 2-cell, mid 2-cell, late 2-cell, 4-cell, 8-cell, 16-cell, early blastocyst, mid blastocyst, and late-blastocyst)[12,13] scRNA-seq datasets were downloaded from the GEO database (see Availability of data and materials). These datasets were generated by the Smart-seq[12] or Smart-seq2[69] (Oocyte Smart-seq, remains, Smart-seq2) experiment strategy and used reference genome mm9. In addition, we used the author-provided reads per kilobase per million mapped reads (RPKM) gene expression matrix for further analysis.

Custom R scripts were used for the pre-implantation scRNA-seq data to calculate the pseudo-bulk expression for an average single cell in each stage. Moreover, the PCC of pseudo-bulk and bulk RNA-seq was calculated using the function "cor" in the R *stats* package. Finally, PCA of pre-implantation scRNA-seq and bulk RNA-seq was performed using the function "prcomp" from the R *stats* package (parameters: center=TRUE, scale.=TRUE). To quantify the presence of batch effects, we conducted the kBET analysis using a $\chi^2$-based test[14]. The kBET analysis assessed whether the data points with different batches in the PCA space were well-mixed within random neighborhoods of a fixed size. The binary test results were averaged to generate an overall rejection rate, which is easily interpretable. A low rejection rate indicates that the batches are well-mixed and that the confounding effects can be neglected. The kBET analysis was performed by R package *kBET* (version: 0.99.6).

Custom R scripts generated the pseudo-bulk expression for each stage from the integrated datasets. Quantile normalization was performed with the function "normalize.quantiles" from R package *preprocessCore* (version 1.48.0). The ligand and receptor gene distribution analysis used the quantile normalized gene expression.

The merged expression matrix was input to *Seurat* (version 3.2.3)[70] to perform t-SNE dimension reduction and draw marker expression by FeaturePlot.

## Ligand and receptor gene expression analysis.

Mouse ligand-receptor gene pairs (2034 pairs) were downloaded from CellTalkDB[16] (http://tcm.zju.edu.cn/celltalkdb/download.php). The ligand and receptor genes were compared with the mm9 protein-coding genes by pseudo-bulk expression with a boxplot using the function "geom_boxplot" from *ggplot2* (version 3.3.5).

We plotted each single-cell gene expression value across stages for a given ligand-receptor gene pair and fit the non-linear

regression curve with the layer function "geom_path" in *ggplot2* (parameters: method = "gam", se = F)

## Metric for cell-cell communication strength.

Cell-cell communication strength was quantified by the interaction score proposed by ref.[17]. The interaction score was defined as follows:

$$Interaction\ score_{ligand,receptor,cell\ type1,cell\ type2}$$
$$= \frac{1}{n_{cell\ type1}}\sum_{i\in cell\ type1}e_{i,ligand} * \frac{1}{n_{cell\ type2}}\sum_{j\in cell\ type2}e_{j,receptor} \quad (1)$$

In which $e_{i,j}$ is the expression value of gene $j$ in cell $i$, and $n_c$ is the number of cells of cell type $c$.

For single-cell autocrine, the interaction score was defined as:

$$Interaction\ score_{receptor,ligand} = \sqrt{e_{receptor} * e_{ligand}} \quad (2)$$

## TimeTalk workflow for identifying eLR.

To identify the eLR from the curated early embryo scRNA-seq, we follow these steps: first, we order the cells according to their trajectories. Then, we identify co-varying eLR and temporal TF (tTFs). Finally, we obtain eLR from co-varying LR through Granger causality analysis with tTFs.

The *monocle2*[15] pipeline was used to construct the trajectory from curated scRNA-seq datasets. First, we used the "differentialGeneTest" function to get differentially expressed genes (cutoff: qval = 0.01) during early embryo development. Then, the differential expressed genes were used to perform DDRTree dimension reduction to reconstruct the development trajectory by the "reduceDimension" function. Finally, pseudotime was calculated by the "orderCells" function.

We arrange the cells by stage and pseudotime. Then, for a given LR, the ordered cells' ligand gene and receptor gene expression values form a two time series. Then, we calculated the Pearson's correlation coefficient (PCC) and used |PCC| = 0.1 as a cutoff to get co-varying LR. LR pairs of PCC >0.1 or PCC < −0.1 is the co-varying LR. The rationale of |PCC| = 0.1 is as follows. We first rank the positively correlated LR in ascending order by PCC, producing a concave curve. The start and endpoints of this curve are labeled as A and B, respectively. We then draw a straight line connecting AB, which is moved to obtain the tangency point C. Next, we connect AC and move the resulting straight line to obtain tangency point D. Below point D, and the curve roughly changes into a linear function. Thus, we select the vertical coordinate of point D as the cutoff value, which is 0.0987, for positive co-varying LR (Supplementary Fig. 7a). Using the same process, the cutoff for the negative co-varying LR is −0.0587 (Supplementary Fig. 7b). To establish the co-varying LR cutoff, we select the maximum value among the positive and negative co-varying LR cutoffs and round to 2 decimal places, resulting in a cutoff of 0.1.

We used the RTN package (version 2.10.1) to reconstruct the transcriptional regulatory network to get temporal TF. In brief, the log10(RPKM + 1) normalized scRNA-seq data and TF list from AnimalTFDB (version 3.0)[71] to build the "tni" object from the RTN package to construct the transcriptional regulatory network. Then, the variable gene and their variance were input as a phenotype to perform master regulator analysis by function "tna.mra". Finally, the output was considered as temporal TFs.

To reduce the computational cost, we clustered the identified tTFs by hierarchical clustering (the agglomeration method was "ward.D2"). After clustering tTFs, we calculate tTFs average expression as the activity of entire TF clusters. Then, we perform Granger causal test with the function "grangertest" from the R package *lmtest* (version 0.9-38). Ultimately, we test the causal relationship between tTFs activity and a given LR pairs

interaction score. Because Granger causal test is an asymmetric test, we test for a given LR pair and a given cluster of tTFs whether there is a Granger-causality relationship from LR to TF or TF to LR.

As we have six clusters of tTFs, we choose the minimal $p$-value to represent the significance of causality from LR to tTFs or tTFs to LR. We used 0.01 as a cutoff to get candidate eLR from co-varying LR. Thus, we consider p_LRtoTF < 0.01 and p_TFtoLR < 0.01 co-varying LR as eLR.

**Validation of eLRs with sets of essential genes, housekeeping genes, and ZGA genes**. The essential gene set was downloaded from the DEG 10 database[24] (http://origin.tubic.org/deg/public/index.php/index). The housekeeping gene set was downloaded from the HRT Atlas v1.0 database[26] (http://www.housekeeping.unicamp.br/). The ZGA gene sets were obtained from Table S5 in ref. [18]. The enrichment ratio of gene set A in a given gene set B was calculated with the following formula:

$$enrichment\ ratio\ of\ A\ in\ B = \frac{|A \cap B|}{|B|} \quad (3)$$

As shown in formula (3), the numerator indicates the number of elements that are common in both set A and set B, while the denominator represents the total number of elements in set B.

For example, the gene set enrichment ratio of essential genes in all eLR genes equals the number of essential genes among all eLRs divided by the number of genes composite of eLRs.

The p-value was calculated by the "phyper" function from R package *stats* (version 3.6.3). For the "phyper" function, we used lower.tail=TRUE to test whether the observed enrichment ratio was higher than the expected enrichment ratio, lower.tail=FALSE to test whether the observed enrichment ratio was lower than the expected enrichment ratio.

**ZGA inhibition RNA-seq data analysis**. ZGA inhibition RNA-seq data were obtained from a study by ref. [30].

The ZGA inhibition RNA-seq dataset included three conditions: early 2-cell, late 2-cell, and 2-cell treated with alpha amanitin (i.e., ZGA-inhibited condition). We treated each condition as a cell type and considered autocrine signaling within each condition.

Two rounds of hierarchical clustering identified the eLR pairs affected by ZGA inhibition. Euclidean distance and complete clustering methods were used for hierarchical clustering. The first step identified the variable eLR pairs, and the second step identified the eLR pairs affected by ZGA inhibition.

We define the delta value to measure the effect of ZGA inhibition on the cell-cell communication strength of LR pairs. The ZGA inhibition RNA-seq dataset included three conditions: early 2-cell, late 2-cell, and 2-cell treated with alpha amanitin (i.e., ZGA-inhibited condition). For a given LR pair with an interaction score IS, the delta value was defined as follows:

$$delta\ value = \left| \begin{array}{l} \left( \log_{10}\left(IS_{2cell\ alph\ aamanitin} + 1\right) - \log_{10}\left(IS_{early\ 2-cell} + 1\right)\right) - \\ \left( \log_{10}\left(IS_{late\ 2-cell} + 1\right) - \log_{10}\left(IS_{early\ 2-cell} + 1\right)\right) \end{array} \right| \quad (4)$$

The formula can be reduced to:

$$delta\ value = \left| \log_{10}\left(IS_{2cell\ alph\ aamanitin} + 1\right) - \log_{10}\left(IS_{late\ 2-cell} + 1\right) \right| \quad (5)$$

We calculated the $p$-value in Supplementary Fig.13 using the "t.test" function from the R package stats (version 3.6.3). To test

whether the delta value of non_eLR was lower than the delta value of eLR, we used alternative = "l" for the "$t$ test" function.

**Blastoid scRNA-seq data analysis**. The scRNA-seq datasets for droplet-based EPS-generated blastoids and natural blastocysts were obtained from a study by ref. [45].

For the EPS-generated blastoid and natural blastocyst data, empty droplets were identified by the function "barcodeRanks" in the R package *DropletUtils* (version 1.6.1, parameters: lower = 100, fit.bounds = NULL, df = 20). After removing the empty droplets, the single-cell gene expression matrix was loaded into R by the function "CreateSeuratObject" in *Seurat* (parameters: min.features = 2000, min.cells = 0). Next, the natural blastocyst and EPS-generated blastoid data were integrated by CCA-based methodology in *Seurat*[70]. First, the anchors for integration were identified by the function "FindIntegrationAnchors" (parameters: dims = 1:20, k.anchor = 5, k.filter = 30). In the next step, the identified anchors were input into the function "IntegrateData" (parameters: dims = 1:20) to obtain integrated data. Finally, the integrated data were processed using the function "ScaleData" (parameters: model.use = "linear", do.scale = TRUE, do.center = TRUE", scale.max = 10, block.size = 1000, min.cells.to.block = 3000), "RunPCA" (parameters: npcs = 30), "RunUMAP" (parameters: reduction = "pca", dims = 1:30, umap.method = "umap-learn"), "FindNeighbors" (parameters: dims = 1:20, reduction = "pca"), and "FindClusters" (parameters: resolution = 0.2). Cell lineages were assigned based on the following markers according to the original research of ref. [45]:

Trophectoderm (TE): *Cdx2, Krt8, Krt18, Ascl2, Tacstd2*;

Inner cell mass or epiblast (ICM/EPI): *Pou5f1, Nanog, Sox2, Esrrb, Sox15*;

Primitive endoderm (PE): *Gata4, Gata6, Sox17, Pdgfra, Col4a* 2C like: Zscan4f, Zscan4c, Zscan4d.

We used Monocle3 to reconstruct trajectories. The integrated UMAP embedding was used to learn trajectory using the function "learn_graph" from the Monocle3 package. Finally, we choose the 2C_like cells as the root cell to calculate psedotime along the trajectory.

**The extended TimeTalk workflow**. The extended TimeTalk workflow including following steps: Firstly, we order cells based on the reconstructed trajectory. Next, we identify the co-varying LR from the two types of cells. Then, we identify the master regulators for each cell type. Finally, we obtain the active LR from the co-varying LR with the master regulators of a particular cell type using Granger's causality.

As *monocle3*[41] can reconstruct complex development trajectories from scRNA-seq datasets, we used *monocle3* to replace *monocle2* to order cells in each cell type.

For a given LR, consider sender cells A expressed ligand(L) and receiver cell B expressed receptor (R). When examining the expression data in ordered cells of both cell type A and cell type B, two time series of unequal (or equal, if A and B are the same types of cell) length were generated. Consequently, it is necessary to calculate the correlation between these two time series. To obtain equal sample points for both cell types, we used the "interWeight" function from the *cellAlign*[42] package to interpolate the expression data along one trajectory (window size of interpolation: winSz = 0.1, number of desired interpolated points, numPts = 200). Using this strategy, we can calculate Spearman's correlation coefficient of interpolated ligand signals calculated from ligand gene expression in sender cells and receptor signals calculated from the receptor gene expression in receiver cells. We used SCC = 0.2 as a cutoff to identify co-varying LR.

We used the function RunALRA from the R package Seurat to perform zero-preserved imputation by the ALRA algorithm[43]. Each cell type's imputed matrix was used to build the transcriptional network and perform master regulator analysis by RTN package[44]. The variable gene along the trajectories identified by the function *graph_test* from the Monocle3 package was used as a phenotype to perform master regulator analysis.

We perform the Granger test to calculate the Granger causality between the gene expression of master TFs from the receiver cell types and the interaction score of co-varying LR pairs from the interpolated pseudotime time series. Meanwhile, we calculate Spearman's correlation coefficient to measure the correlation between the activity of co-varying LR and master TFs. We used 0.01 as a cutoff to test whether the given LR and a master TF have Granger causality. Thus, for a given LR, it will have many master TFs passed the Granger test and have many different correlation values. We choose the maximum (noted as SCC_ens) as the final correlation value to measure the correlation of LR with the TFs activity. We used $SCC\_ens > 0.8$ or $SCC\_ens < -0.8$ to get active LR.

**Co-evolution analysis**. We used the HomoloGene database (https://www.ncbi.nlm.nih.gov/homologene) to retrieve homologous ligand and receptor genes in four mammalian species *Mus Musculus* (mouse), *Rattus norvegicus* (rat), *Bos Taurus* (cattle), and *Homo sapiens* (human). Through this approach, 1119 homologous genes were obtained, which included 922 ligand-receptor pairs, of which 208 were eLR pairs, and 714 were non-eLR pairs. The ratio of nonsynonymous and synonymous substitution rates (Ka/Ks) for each gene was calculated using the following steps: amino acid sequence alignments were performed using the R package msa (version 1.28.0)[72] and ClusterW algorithm[73], followed by codon alignment using the codon_aln function by PAL2NAL algorithm[74] in the R package orthologr (version 0.4.0)[75]. The KaKs ratio was then calculated using the dnastring2kaks function in the R package MSA2dist (version 1.0.0). Finally, co-evolution trends were evaluated by computing the Pearson correlation coefficient (PCC) of the Ka/Ks ratio for each ligand-receptor pair in each pairwise species comparison.

**Statistics and reproducibility**. All statistical analyses were performed using R (version 3.6.3). Heatmaps were generated by the R package *ComplexHeatmap* (version 2.9.4)[76]. Graphs of statistical results were generated by the R package *ggplot2* (version 3.3.5). GO analysis and KEGG analysis were performed by the R package *clusterProfiler* (version 3.14.3)[77].

A collection of publicly available pre-implantation single-cell RNA-seq datasets were used to analyze the different stages of early embryo development. These stages include the oocyte (3 cells), zygote (4 cells), early 2-cell (8 cells), mid 2-cell (12 cells), late 2-cell (10 cells), 4-cell (14 cells), 8-cell (28 cells), 16-cell (50 cells), early blastocyst (43 cells), mid blastocyst (60 cells), and late blastocyst (30 cells). In total, there were 262 cells analyzed to ensure that all stages were represented in the scRNA-seq datasets. In Supplementary Note 1, we tested the impact of interpolation strategies on Granger causality conclusions by using two different time series models with theoretical causality. To determine the false positive ratio of Granger causality that resulted from interpolation, we conducted ten rounds of simulations for each model. In each round of simulation, we generated simulation data and performed the Granger test on the interpolated data 1000 times. Since each model has a theoretical Granger causality, the false positive ratio was calculated for each round.

In order to make sure that our results can be replicated, we have included the source data for the critical analysis of our manuscript as Supplementary Data. Additionally, we have uploaded our code and extensive data to Figshare (https://figshare.com/projects/TimeTalk_CB_manuscript_code_data/174498) to make it easier for others to reproduce our analyses.

**Reporting summary**. Further information on research design is available in the Nature Portfolio Reporting Summary linked to this article.

## Data availability

The publicly available data on early embryo development used in this paper were obtained from the Gene Expression Omnibus (GEO) database (https://www.ncbi.nlm.nih.gov/geo/). The mouse oocyte scRNA-seq data were obtained from a study by ref. [12] with accession number GSE38495. The mouse scRNA-seq data covering the zygote stage to the late blastocyst stage were obtained from a study by ref. [13] with accession number GSE45719. The mouse bulk RNA-seq was obtained from studies by Wu et al. with accession numbers GSE66582. The mouse ZGA inhibition RNA-seq data were obtained from a study by ref. [30] with accession number GSE71434. The EPS-generated blastoid scRNA-seq data and natural blastocyst scRNA-seq data were obtained from a study by ref. [45] with accession number GSE135701. The processed sequencing data used in this article has been deposited in the Figshare repository (https://doi.org/10.6084/m9.figshare.23850324.v7)[78]. Source data underlying Figs. 2f, g, and 3a–d, and Supplementary Fig. 13 are provided in Supplementary Data 1. Source data for Fig. 4a–f and Supplementary Fig. 16a–c are provided in Supplementary Data 2. Source data for Fig. 5c–e are provided in Supplementary Data 3.

## Code availability

All custom scripts required to reproduce all results reported in this manuscript have been deposited in Figshare repository (https://doi.org/10.6084/m9.figshare.23895780.v6)[79]. The source code for TimeTalk R package can be accessed on Github (https://github.com/ChengLiLab/TimeTalk)[80]. Furthermore, we have also deposited the TimeTalk R package in Zenodo (https://doi.org/10.5281/zenodo.8271645)[81].

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

## Acknowledgements

We are grateful to the researchers who produced the publicly available data used in this study. In addition, we are grateful to Prof. Jin Zhang at Zhejiang University, Prof. Dahai Zhu at Peking Union Medical College, Ming Yang of Prof. Peng Du'lab at Peking University,Yang Ding of Prof. Xiaochen Bo's lab at Institute of Health Service and Transfusion Medicine, Molong Qu and Liang Xiong at Prof. Hongkui Dengs' lab at Peking University,Yuting Liu, Zihan Xu, Yulin Lyu, Yaqi Wang and Qing Fang of Prof. Cheng Li's Lab at Peking University for providing sound advice regarding this research. This work was supported by the National Natural Science Foundation of China (32288102 and 32025006 to C.L., 62173338 to H.C., 61873276 to X.B.) and the National Key Research and Development Program of China (2021YFA1100300 to C.L.), and the Beijing Nova Program of Science and Technology (20220484198 to H.C.). Part of the data analysis was performed on the High-Performance Computing Platform of the Center for Life Sciences, Peking University.

## Author contributions

L.W., H.C., H.L., X.B., and C.L. conceived and designed the study. L.W. and S.M. performed the background investigation. L.W. performed all the bioinformatics analyses. L.W. and Y.Z. created the scientific illustrations. L.W., Y.Z., Y.S., S.M., H.L. and H.C. wrote the manuscript.

## Competing interests

The authors declare no competing interests.
