## [Peer Review File · Communications Biology]

Reviewers' comments:

Reviewer #1 (Remarks to the Author):

In this manuscript, the authors introduce TimeTalk, a computational approach that uses time-course single-cell RNA-seq data to identify ligand-receptor pairs for studying cell-cell communications in early embryo development. They applied TimeTalk to predict hundreds of early embryo development-related ligand-receptor pairs (eLRs). Cross-referencing with orthogonal information suggests that the predicted eLRs may have important functions in embryonic development. Overall, TimeTalk can be a useful tool for studying cell-cell communications in a temporal dynamic process. However, there are a number of issues that need to be addressed to make the study more convincing.

Major issues:

1. The MII-oocyte (E0) scRNA-seq data and E1-E4.0 scRNA-seq data are obtained from two separate studies. Here the temporal changes of a gene between E0 and E1-E4.0 can be confounded with batch differences. How will this potential confounding influence the conclusions in this paper?
2. In Figure 1a, there is no bulk RNA-seq for MII-oocyte. Where does the bulk MII oocyte in Figure 1b come from?
3. Strictly speaking, Granger test does not infer causal relationship. It only evaluates the usefulness of one variable to forecast another. Using statements such as "make causal inference" in the paper is misleading.
4. In line 278-280, pseudotime of two cell types are aligned by interpolating equal sample points using CellAlign. This assumes that the pseudotime of two cell types with unequal length of time-series can be meaningfully compared after normalizing their time length by CellAlign. Are the conclusions from the Granger test sensitive to this assumption? If the pseudotime of two cell types cannot be correctly aligned under this assumption, then comparing the pseudotemporal curves of the two cell types may be misleading. Since Granger test uses genes' temporal expression curves to infer Granger causality, the test results may change if the temporal relationship between two curves (e.g. the temporal order of peaks of the two curves) from two cell types is changed. A natural question is that if the above assumption regarding the alignment is not true, can one still obtain the same Granger causality conclusions?
5. Line 238, the statement "the involvement of the identified eLR pairs in early embryo development was confirmed" needs to be toned down. The in silico analyses only provide indirect evidence suggesting that the eLR pairs are involved in early embryo development. They do not provide direct evidence to "confirm" that the 430 identified eLR pairs are all involved in this process.
6. It is unclear which figures are used to support statements in lines 246-251. The authors cite Figure 4a, but the figure legend does not provide any information to support the statement such as "Cluster1 eLRs have backward relationship with C1 TFs..."
7. The formula for Interaction score in line 395 does not make sense: ligand does not appear anywhere in the right hand side of the equation. $n_{\text{cell type 1}}$ and $n_{\text{cell type 2}}$ on the right hand side are also used incorrectly.
8. Line 414: what is the rationale for using $PCC=0.1$ as a cutoff to get co-vary LR? In Figure 2b, the PCC between the known ligand-receptor pair Fgf4-Fgfr2 is negative, so it won't pass the $PCC=0.1$ cutoff?
9. Fig 3e: as a negative control, does ZGA inhibition alter the cell communication strength of LR pairs

that are not eLR?

Minor issues:

There are numerous grammar errors. A few examples below:

1. Line 27-28 (Abstract): "To make TimeTalk can be applied to study paracrine" is grammatically incorrect.
2. Line 58: "The high expression Nanog of progenitor cells in ICM [expression] high levels of Fgf4".
3. Line 69: "we developed and applied a computational framework named TimeTalk [utilized] temporal series information [though] study the dynamics of autocrine within the embryos"

There are more such errors throughout the manuscript. The authors should carefully proofread the manuscript and fix them.

Reviewer #2 (Remarks to the Author):

In the present manuscript, Wang et.al. discussed the development "TimeTalk", a tool developed to study cell to cell communication in the embryonic developmental process. Early stages of embryonic development are a complex process of self-assembly where deconvolution of the events leading to this orchestrated process is extremely important.

Authors by developing this tool "TimeTalk" have aimed to resolve the events and investigated how cells communicate during the process. Deployment of this computational framework would be in general helpful to probe into the cell-cell communication and the series of events associated with other developmental processes and in disease contexts where this communication and series of events are impaired. The article is well written, and the message is clear. But in many cases contextualization or introduction of some relevant terminologies, are missing; this could be a big hindrance for a broader readership. Also, there are segments with grammatical errors, more precisely sentence framing errors. These need to be rectified. Given the strength of the tool developed and the likely impact it would have in deciphering cell-cell communication, I would consider the article for publication upon addressing the issues I highlighted.

Here are few specific major concerns which need to be addressed:

Early embryonic development is a multi-stepped process and the "TimeTalk" framework aims to resolve the events by probing into cell-cell communication. But the authors did not include any proper detailing of the embryonic development, save some very brief outline in the introduction. A little more details would be helpful for the broader readership.

The objective sentence in the introduction suddenly brings in the relevance of time-course scRNA-seq datasets; this is hard to comprehend in the introduction section and hence needs some context and a brief introduction.

There are sentence framing errors in Introduction (and in other sections of the article).

Authors while citing the interplay between tTF and e-LR mentioned the LR, and TF formed gene regulator network. Authors need to elaborate on this and mention how it is relevant.

It would be helpful for the readers if authors justify and elaborate on how cluster 4 and 6 enriched the Hippo signaling pathway.

Authors in the discussion should also discuss the scope of the TimeTalk framework in context of other cell-cell communication and in the context of altered states, like diseases.

It would be interesting if authors discuss the ligand receptor interaction in the context of co-evolution

of ligand and receptor.

Reviewer #3 (Remarks to the Author):

In this article, authors proposed a computational frame to track the cell to cell communication through ligand-receptors binding, leads to cell fate during early embryonic development.

The proposed framework is piece of fresh concept addressing an interesting biological questions. The outcomes look promising and relevant enough. Though there are few major concerns major issue:

1. The language of the article is not elucidate enough.
2. Use acronyms carefully.
3. Figure legends should be more explanatory
4. The flow of the summary is little disconnected

Reviewers' comments

Reviewer #1 (Remarks to the Author):

In this manuscript, the authors introduce TimeTalk, a computational approach that uses time-course single-cell RNA-seq data to identify ligand-receptor pairs for studying cell-cell communications in early embryo development. They applied TimeTalk to predict hundreds of early embryo development-related ligand-receptor pairs (eLRs). Cross-referencing with orthogonal information suggests that the predicted eLRs may have important functions in embryonic development. Overall, TimeTalk can be a useful tool for studying cell-cell communications in a temporal dynamic process. However, there are a number of issues that need to be addressed to make the study more convincing.

Major issues:

1. The MII-oocyte (E0) scRNA-seq data and E1-E4.0 scRNA-seq data are obtained from two separate studies. Here the temporal changes of a gene between E0 and E1-E4.0 can be confounded with batch differences. How will this potential confounding influence the conclusions in this paper?
2. In Figure 1a, there is no bulk RNA-seq for MII-oocyte. Where does the bulk MII oocyte in Figure 1b come from?
3. Strictly speaking, Granger test does not infer causal relationship. It only evaluates the usefulness of one variable to forecast another. Using statements such as "make causal inference" in the paper is misleading.
4. In line 278-280, pseudotime of two cell types are aligned by interpolating equal sample points using CellAlign. This assumes that the pseudotime of two cell types with unequal length of time-series can be meaningfully compared after normalizing their time length by CellAlign. Are the conclusions from the Granger test sensitive to this assumption? If the pseudotime of two cell types cannot be correctly aligned under this assumption, then comparing the pseudotemporal curves of the two cell types may be misleading. Since Granger test uses genes' temporal expression curves to infer Granger causality, the test results may change if the temporal relationship between two curves (e.g. the temporal order of peaks of the two curves) from two cell types is changed. A natural question is that if the above assumption regarding the alignment is not true, can one still obtain the same Granger causality conclusions?
5. Line 238, the statement "the involvement of the identified eLR pairs in early embryo development was confirmed" needs to be toned down. The in silico analyses only provide indirect evidence suggesting that the eLR pairs are involved in early embryo development. They do not provide direct evidence to "confirm" that the 430 identified eLR pairs are all involved in this process.

6. It is unclear which figures are used to support statements in lines 246-251. The authors cite Figure 4a, but the figure legend does not provide any information to support the statement such as "Cluster1 eLRs have backward relationship with C1 tTFs..."

7. The formula for Interaction score in line 395 does not make sense: ligand does not appear anywhere in the right hand side of the equation. $n_{\text{cell type 1}}$ and $n_{\text{cell type 2}}$ on the right hand side are also used incorrectly.

8. Line 414: what is the rationale for using $PCC=0.1$ as a cutoff to get co-vary LR? In Figure 2b, the PCC between the known ligand-receptor pair Fgf4-Fgfr2 is negative, so it won't pass the $PCC=0.1$ cutoff?

9. Fig 3e: as a negative control, does ZGA inhibition alter the cell communication strength of LR pairs that are not eLR?

Minor issues:

There are numerous grammar errors. A few examples below:

1. Line 27-28 (Abstract): "To make TimeTalk can be applied to study paracrine" is grammatically incorrect.

2. Line 58: "The high expression Nanog of progenitor cells in ICM [expression] high levels of Fgf4".

3. Line 69: "we developed and applied a computational framework named TimeTalk [utilized] temporal series information [though] study the dynamics of autocrine within the embryos"

There are more such errors throughout the manuscript. The authors should carefully proofread the manuscript and fix them.

Reviewer #2 (Remarks to the Author):

In the present manuscript, Wang et.al. discussed the development "TimeTalk", a tool developed to study cell to cell communication in the embryonic developmental process. Early stages of embryonic development are a complex process of self-assembly where deconvolution of the events leading to this orchestrated process is extremely important. Authors by developing this tool "TimeTalk" have aimed to resolve the events and investigated how cells communicate during the process. Deployment of this computational framework would be in general helpful to probe into the cell-cell communication and the series of events associated with other developmental processes and in disease contexts where this communication and series of events are impaired. The article is well written, and the message is clear. But in many cases contextualization or introduction of some relevant terminologies, are missing; this could a big hindrance for a broader readership. Also, there are segments with grammatical errors, more precisely

sentence framing errors. These need to be rectified. Given the strength of the tool developed and the likely impact it would have in deciphering cell-cell communication, I would consider the article for publication upon addressing the issues I highlighted.

Here are few specific major concerns which need to be addressed:

Early embryonic development is a multi-stepped process and the "TimeTalk" framework aims to resolve the events by probing into cell-cell communication. But the authors did not include any proper detailing of the embryonic development, save some very brief outline in the introduction. A little more details would be helpful for the broader readership.

The objective sentence in the introduction suddenly brings in the relevance of time-course scRNA-seq datasets; this is hard to comprehend in the introduction section and hence needs some context and a brief introduction.

There are sentence framing errors in Introduction (and in other sections of the article).

Authors while citing the interplay between tTF and e-LR mentioned the LR, and TF formed gene regulator network. Authors need to elaborate on this and mention how it is relevant.

It would be helpful the for the readers if authors justify and elaborate on how cluster 4 and 6 enriched the Hippo signaling pathway.

Authors in the discussion should also discuss the scope of the TimeTalk framework in context of other cell-cell communication and in the context of altered states, like diseases.

It would be interesting if authors discuss the ligand receptor interaction in the context of co-evolution of ligand and receptor.

Reviewer #3 (Remarks to the Author):

In this article, authors proposed a computational frame to track the cell to cell communication through ligand-receptors binding, leads to cell fate during early embryonic development.

The proposed framework is piece of fresh concept addressing an interesting biological questions. The outcomes look promising and relevant enough. Though there are few major concerns

major issue:

1. The language of the article is not elucidate enough.

2. Use acronyms carefully.
3. Figure legends should be more explanatory
4. The flow of the summary is little disconnected

Point-to-point response

Reviewer #1

In this manuscript, the authors introduce TimeTalk, a computational approach that uses time-course single-cell RNA-seq data to identify ligand-receptor pairs for studying cell-cell communications in early embryo development. They applied TimeTalk to predict hundreds of early embryo development-related ligand-receptor pairs (eLRs). Cross-referencing with orthogonal information suggests that the predicted eLRs may have important functions in embryonic development. Overall, TimeTalk can be a useful tool for studying cell-cell communications in a temporal dynamic process. However, there are a number of issues that need to be addressed to make the study more convincing.

We greatly appreciate the positive feedback from the reviewer regarding the benefits of TimeTalk. We would also like to express our sincere gratitude to reviewer 1 for providing insightful and constructive suggestions and criticisms of our study. In response to reviewer 1's valuable advice, we have carefully incorporated all comments into the revised version of our research to enhance its quality.

Major issues:

R1-1. The MII-oocyte (E0) scRNA-seq data and E1-E4.0 scRNA-seq data are obtained from two separate studies. Here the temporal changes of a gene between E0 and E1-E4.0 can be confounded with batch differences. How will this potential confounding influence the conclusions in this paper?

Response:

We appreciate the concerns raised by the reviewer. Although the E0 scRNA-seq datasets (GEO accession: GSE38495)¹ and E1-E4.0 scRNA-seq datasets (GEO accession, GSE45719)² were obtained from two different studies; however, they were both produced by Rickard Sandberg group with the smart-seq protocol. Indeed, the two datasets were merged without batch correction by Deng et.al². to study the temporal changes during early embryo development.

Indeed, in our original manuscript, we have compared the bulk RNA-seq and scRNA-seq datasets to confirm that each stage of data captures the characteristics of corresponding development stages (Fig 1b. see below).

To quantify the presence of batch effects, we conducted the kBET analysis using a χ^2 -based test³. The kBET analysis assessed whether the data points with different batches in the PCA space were well-mixed within random neighborhoods of a fixed size. The binary test results were averaged to generate an overall rejection rate, which is easily interpretable. A low rejection rate indicates that the batches are well-mixed and that the confounding effects can be neglected. The results of our kBET analysis, as displayed in Supplementary Fig. 3 (refer below), indicate that the observed rejection rate is low. This suggests that the batch effects can be disregarded, as mixing data points with different batches in the PCA space is satisfactory. We have incorporated this portion of results into the "Curation of early-embryo development single-cell RNA-seq data sets for studying cell-cell communication" subsection within the Results section of the revised manuscript. Furthermore, we have included relevant details in the Methods section of the revised manuscript.

R1-2. In Figure 1a, there is no bulk RNA-seq for MII-oocyte. Where does the bulk MII oocyte in Figure 1b come from?

Response:

We want to express our apologies for any confusion arising from an ambiguous description provided, which may have led to misinterpretation of the information presented. Similar to the other bulk RNA-seq datasets in Figure 1a, the bulk MII oocyte data was obtained from identical GEO datasets and can be recognized by its corresponding accession ID GSE66582⁴. Additionally, we have corrected Figure 1a to ensure accuracy.

R1-3. Strictly speaking, Granger test does not infer causal relationship. It only evaluates the usefulness of one variable to forecast another. Using statements such as "make causal inference" in the paper is misleading.

Response:

We express our gratitude to the reviewer for bringing to our attention the issue raised in the manuscript. We concur with the reviewer that the Granger test evaluates the ability of one variable to predict another, which is commonly referred to as Granger causality. Although Granger causality is related to causal relationships, it does not imply actual causality. We apologize for any confusion caused by our imprecise use of the term "causal

inference" in the paper. We have carefully reviewed and revised the manuscript according to the reviewer's suggestions, replacing "make causal inference" with "calculate the Granger causality."

R1-4. In line 278-280, pseudotime of two cell types are aligned by interpolating equal sample points using CellAlign. This assumes that the pseudotime of two cell types with unequal length of time-series can be meaningfully compared after normalizing their time length by CellAlign. Are the conclusions from the Granger test sensitive to this assumption? If the pseudotime of two cell types cannot be correctly aligned under this assumption, then comparing the pseudotemporal curves of the two cell types may be misleading. Since Granger test uses genes' temporal expression curves to infer Granger causality, the test results may change if the temporal relationship between two curves (e.g. the temporal order of peaks of the two curves) from two cell types is changed. A natural question is that if the above assumption regarding the alignment is not true, can one still obtain the same Granger causality conclusions?

Response:

We want to extend our sincere appreciation to the reviewers for their invaluable feedback regarding the sensitivity and reliability of Granger causality concerning the interpolated expression of two cell types. In response to these concerns, we have taken several steps to address the issues raised, including conducting additional analyses and providing comprehensive clarifications. The following outlines the actions taken under the reviewers' suggestions.

- (1) We want to provide clarification regarding our hypotheses. First, as the reviewer points out, the updated version of TimeTalk assumes that the pseudotime of two cell types with varying lengths of time series can be compared meaningfully after interpolating with CellAlign. In addition, we have analyzed the expression patterns of two cell types with the assumption that their participation in intercellular communication, mediated by specific ligand receptors, could potentially connect the gene expression dynamics of ligands and receptors. Moreover, this cell-cell communication between the two cell types may also contribute to the causal relationship with the expression dynamics of transcription factors. Consequently, instead of performing a downstream trajectory alignment from CellAlign to assess timing differences and similarities between the two cell types, we first screen co-varying ligand-receptor pairs using Spearman correlation coefficients. This step involves utilizing the interpolated ligand gene expression vectors in one cell type and the receptor gene expression in another to calculate the correlation coefficient.
- (2) We agree with the reviewer's suggestions that we should test whether the conclusion from TimeTalk is robust with different parameters of the interpolation procedure and the following Granger causal test procedures. In TimeTalk, the interpolation and following Granger causal test procedures rely on three parameters: window size of the interpolation ($winsz$), number of desired interpolated points ($numPts$), and the order of lags to include the auxiliary regression (lag). In our manuscript, we utilized the parameter values of $winsz=0.1$, $numPts=200$, and $lags=1$ to identify 311 ligand-

receptor mediated cell-cell communications between the epiblast and primitive endoderm in a blastocyst (Supplementary Table 2). To further investigate the impact of parameter choices on inferred ligand-receptor interactions, we focused specifically on intercellular communication between the epiblast and primitive endoderm, using the original parameter combination as a reference for the 311 ligand-receptor pairs. We then varied parameter values within specific ranges and calculated the ratio of recalled reference ligand-receptor pairs in the output ligand-receptor pairs with varied parameters to assess the robustness of TimeTalk. As we have three parameters, first, we only varied one parameter from the original parameters' combinations (varied one parameter of the combination winsz=0.1, numPts = 200, lags =1, and fixed the other). Although TimeTalk appears to be highly influenced by the parameter "winsz" (Supplementary Fig. 19a), it remains comparatively unaffected by the "numPts" and "lags" parameters (Supplementary Fig. 19b, c below). Moreover, it remains relatively robust when using the three critical parameters within certain ranges (Supplementary Fig. 19 d). We have incorporated this portion of results into the "TimeTalk workflow can be updated to decipher cell-cell communication in blastoid" subsection within the Results section of the revised manuscript.

(3) We appreciate the reviewer's suggestion reminding us to check the reliability of the causality conclusion deduced by the interpolation strategy. We validate this in the real data and simulated datasets.

- In the real data, we confirmed that TimeTalk could accurately infer existing causality as reported in the literature.

By analyzing the result of TimeTalk, we found a feedback relationship between *Fgf4-Fgfr2* and *Gata6* in cell-cell communication between the epiblast and primitive endoderm ($p_{LR_to_TF} = 0.0184$, $p_{TF_to_LR} = 4.74 \times 10^{-5}$, p-value were calculated by granger causal test Supplementary Fig. 19 e, see below). This result is consistent with previous research⁵ that demonstrated that *Nanog* in the epiblast upregulates the expression of *Fgf4*. The secreted *Fgf4* from epiblast induced *Fgf4-Fgfr2* interaction in primate endoderm cells to release *Gata6* expression through ERK signaling.

then potentiates the upregulation of *Fgfr2*, and the feedback loop reinforces a primitive endoderm fate. We also confirmed that the interpolation result could suggest the *Nanog* upregulates *Fgf4* expression, but *Fgf4* cannot regulate *Nanog* expression in the epiblast (SCC=0.640, $p_{L_to_TF} = 0.657$, $p_{TF_to_L}=0.025$, p-value were calculated by granger causal test, SCC means Spearman correlation coefficient, Supplementary Fig. 19 f, see below).

Additionally, the interpolation result could confirm the feedback relationship between *Fgfr2* and *Gata6* (SCC = 0.274, $p_{R_to_TF}=1.15 \times 10^{-10}$, $p_{TF_to_LR}=9.19 \times 10^{-5}$, p-value were calculated by the Granger causal test, Supplementary Fig 19 g, see below). However, we observed that the interpolated curves of the interaction scores of *Fgf4-Fgfr2* and the interpolated curves of *Gata6* expression were negatively correlated. While these findings must be further confirmed, they suggest a complex quantitative relationship between ligand-receptors' interaction and transcription factor activity in receiver cells. Importantly, as high-throughput and high temporal resolution technologies become increasingly available, it will be critical to reconsider traditional notions of positive/negative regulation between these two biological entities. We have incorporated this portion of results into the "TimeTalk workflow can be updated to decipher cell-cell communication in blastoid" subsection within the Results section of the revised manuscript.

- As high temporal resolution data was unavailable, we utilized two models with theoretical Granger causality to create simulation datasets. These datasets were then employed to examine how the interpolation strategy might impact the conclusion of Granger causality.

a) Based on the definition of Granger causality for two time series X_t and Y_t , if X_t is the Granger cause of Y_t , then

$$\sigma^2(Y_t | \bar{Y}_t, \bar{X}_t) < \sigma^2(Y_t | \bar{Y}_t)$$

b) The first time series model is two independent white noise:

$$Y_t = \epsilon_t, X_t = \eta_t, \epsilon_t \sim WN(0,1), \eta_t \sim WN(0,1) \quad (1)$$

As X_t, Y_t are independent, we can deduce that

$$\sigma^2(Y_t | \bar{Y}_t, \bar{X}_t) = \sigma^2(Y_t | \bar{Y}_t) = \sigma^2(Y_t) = 1$$

and $\sigma^2(X_t | \bar{X}_t, \bar{Y}_t) = \sigma^2(X_t | \bar{X}_t) = \sigma^2(X_t) = 1$, therefore, X_t is not the Granger cause of Y_t and Y_t is not the Granger cause of X_t .

According to model (1), we can generate the simulation time series datasets with length N (we choose $N=200$). We sample each time series with length M (we choose $M=50$). Based on our previous investigation, $winsz$ is the most influential parameter in interpolation. Therefore we varied $winsz$ and tested how $winsz$ influenced the imputed time series curve by interpolation strategy. We found that for model (1), the imputed datasets generated by small $winsz$ can preserve the conclusion that neither X_t or Y_t is causally related to the other (Supplementary Fig. 20 a-c, see below). However, larger $winsz$ can lead to false conclusions (Supplementary Fig. 20 d-f, see below).

To prevent arbitrariness, we conducted ten simulation rounds, each with 1000 replications of the above procedure. The results showed that a high ratio of false Granger causality still holds for larger $winsz$ (Supplementary figure 20 g-l, see below, p-value were calculated by the left Wilcox test).

c) The second time series model is two independent white noise:

$$Y_t = X_{t-1} + \epsilon_t, X_t = \eta_t + 0.5\eta_{t-1}, \epsilon_t \sim WN(0,1), \eta_t \sim WN(0,1) \quad (2)$$

We can calculate that (the details can be found at

https://www.math.pku.edu.cn/teachers/lidf/course/fts/ftsnotes/html/_ftsnotes/causal.html)

$$\sigma^2(Y_t | \bar{Y}_t, \bar{X}_t) = 1, \sigma^2(Y_t | \bar{Y}_t) \approx 2.1328$$

$$\text{and } \sigma^2(X_t | \bar{X}_t, \bar{Y}_t) = \sigma^2(X_t | \bar{X}_t) = 1,$$

therefore, X_t is the Granger cause of Y_t and Y_t is not the Granger cause of X_t .

According to model (2), we can generate the simulation time series datasets with length N (we choose N=200). We replicated the procedure described in (b). We sample each time series with length M (we choose M=50). Contrary to model (1), for model (2), the imputed datasets generated by small winsz cannot preserve the conclusion X_t is the Granger cause of Y_t and Y_t is not the Granger causal of X_t . (Supplementary Fig. 21 a-c, see below).

When the value of the "winsz" parameter is increased, the imputed datasets can effectively maintain the causality relationship that exists between X_t and Y_t , but not vice versa. To prevent arbitrariness, we also conducted ten simulation rounds, each involving 1000 replications of the above procedure. The high ratio of false Granger causality between Y_t and X_t still holds for larger winsz (Supplementary Fig. 21 g-l, see above).

We have added Supplementary Note 1 to incorporate these simulation results.

(d) According to our simulation results, the Granger causality conclusion is affected by choice of parameter winsz. Thus, as presented in this manuscript, we acknowledge the need for additional methods to validate the inferred potential regulatory relationship between ligand-receptor interactions and transcription factors. Regrettably, the scarcity of high temporal resolution datasets hinders our ability to address this issue. Despite this limitation, we have acknowledged in the Discussion section of the manuscript that our causal analysis serves as a helpful reference, and we intend to improve our approach in the future with more accurate data. Furthermore, we advise users in the TimeTalk documentation to conduct additional data analysis and experiment with various parameter combinations to achieve more dependable results when utilizing TimeTalk for cell communication analysis.

R1-5. Line 238, the statement "the involvement of the identified eLR pairs in early embryo development was confirmed" needs to be toned down. The in silico analyses only provide indirect evidence suggesting that the eLR pairs are involved in early embryo development. They do not provide direct evidence to "confirm" that the 430 identified eLR pairs are all involved in this process.

Response:

We sincerely appreciate the reviewer for taking the time to provide us with their valuable comments on our manuscript. We sincerely apologize for using the term "confirmed" to describe the involvement of the identified eLR pairs in early embryo development, as we now realize that this term overstates the evidence provided by in silico analyses. We appreciate the reviewer's clarification that in silico analyses only provide indirect evidence that suggests the eLR pairs are involved in early embryo development.

R1-6. It is unclear which figures are used to support statements in lines 246-251. The authors cite Figure 4a, but the figure legend does not provide any information to support the statement, such as "Cluster1 eLRs have a backward relationship with C1 tTFs..."

Response:

We apologize for this inconvenience. In addition, we amended the figure legend of Figure 4a to include information that supports the statement about the backward relationship between Cluster1 eLRs and C1 tTFs:

"Heatmap of the interaction scores of all identified 430 eLRs can be grouped into 6 clusters. In the row annotation of the heatmap, the "cluster" column illustrates the classification of the eLR, C1, C2, C3, C4, C5, and C6 columns illustrate the relationship of each eLR to the C1, C2, C3, C4, C5, C6 regulons. In the column annotation of the heatmap, the "Stage" row illustrates the development stage of every single cell, the "Pseudotime" row illustrates the pseudotime, C1, C2, C3, C4, C5, C6 row illustrates the activity of every single cell."

Thank you for helping us improve the quality of our manuscript.

R1-7. The formula for Interaction score in line 395 does not make sense: ligand does not appear anywhere in the right hand side of the equation. $n_{\text{cell type 1}}$ and $n_{\text{cell type 2}}$ on the right hand side are also used incorrectly.

Response:

We apologize for the oversight and appreciate the helpful feedback from the reviewer. Unfortunately, there was a typographical error in the formula for the Interaction score on line 395, and we regret any confusion this may have caused. The correct formula is as follows:

$$\text{Interaction score}_{\text{ligand, receptor, cell type 1, cell type 2}} = \frac{1}{n_{\text{cell type 1}}} \sum_{i \in \text{cell type 1}} e_{i, \text{ligand}} * \frac{1}{n_{\text{cell type 2}}} \sum_{j \in \text{cell type 2}} e_{j, \text{receptor}}$$

In which $e_{i,j}$ is the expression value of gene j in cell i , and n_c is the number of cells of cell type c .

We revised our manuscript to reflect the correct formula.

R1-8. Line 414: what is the rationale for using PCC=0.1 as a cutoff to get co-vary LR? In Figure 2b, the PCC between the known ligand-receptor pair Fgf4-Fgfr2 is negative, so it won't pass the PCC=0.1 cutoff?

Response:

We want to express our gratitude to the reviewer for bringing to our attention the need to clarify the procedure for obtaining co-vary LR. We apologize for any confusion that may have arisen in the original manuscript. The following steps provide a detailed explanation of how we obtain the co-vary LR.

First, we apologize for the oversight in line 414. Indeed, we use $|PCC| = 0.1$ as a cutoff to get co-vary LR LR pairs of $PCC > 0.1$ or $PCC < -0.1$ as the co-vary LR. Thus, Fgf4-Fgfr2 passed the cutoff.

Second, the rationale of $|PCC| = 0.1$ is as follows.

The above figure illustrates how we obtain the positively correlated co-vary LR cutoff (Supplementary Fig. 7a). We first rank the positively correlated LR in ascending order by PCC, producing a concave curve. The start and endpoints of this curve are labeled as A and B, respectively. We then draw a straight line connecting A.B., which is moved to obtain the tangency point C. Next, we connect A.C. and move the resulting straight line to obtain tangency point D. The curve below point D exhibits a nearly-linear trend. Therefore, we have selected the vertical coordinate of point D, which is 0.0987, as the cutoff value for positive co-vary LR.

Using the same process, the cutoff for the negative co-vary LR is -0.0587. (Supplementary Fig. 7b). To establish the co-vary LR cutoff, we select the maximum value among the positive and negative co-vary LR cutoffs and round to 2 decimal places, resulting in a cutoff of 0.1.

We have integrated these results into the Methods portion of the revised manuscript to provide a more comprehensive explanation of the process for obtaining co-vary LR.

R1-9. Fig 3e: as a negative control, does ZGA inhibition alter the cell communication strength of LR pairs that are not eLR?

Response:

The reviewer provided constructive feedback regarding our original manuscript, suggesting that we investigate the effect of ZGA inhibition on the cell-cell communication strength of LR pairs that are not eLR. To address this concern, we performed additional analysis.

We define the delta value to measure the effect of ZGA inhibition on the cell-cell communication strength of LR pairs. The ZGA inhibition RNA-seq dataset included three conditions: early 2-cell, late 2-cell, and 2-cell treated with alpha amanitin (i.e., ZGA-inhibited condition). For a given LR pair with the interaction score IS, the delta value was defined as follows:

$$\text{delta value} = |(\log_{10}(IS_{2\text{cell alpha amanitin}} + 1) - \log_{10}(IS_{\text{early 2-cell}} + 1)) - (\log_{10}(IS_{\text{late 2-cell}} + 1) - \log_{10}(IS_{\text{early 2-cell}} + 1))|$$

The formula can be reduced to:

$$\text{delta value} = |\log_{10}(IS_{2\text{cell alpha amanitin}} + 1) - \log_{10}(IS_{\text{late 2-cell}} + 1)|$$

Our analysis revealed that the delta value for eLR was lower than that of LR pairs that are not eLR (Supplementary Fig. 13, p-value were calculated by t-test, see below). These results demonstrate that the impact of ZGA inhibition is more notable in non-eLR pairs as compared to eLR pairs. This result suggests that eLR pairs possess additional regulation mechanisms beyond ZGA.

We have integrated these results into the Methods and Results portion of the revised manuscript to provide a more comprehensive exploration of how ZGA inhibition.

Minor issues:

R1-10 There are numerous grammar errors. A few examples below:

Response:

Thank the reviewer for altering us to this issue. We apologize for any inconvenience caused by the grammar errors present in the manuscript. Taking the valuable feedback provided

by the reviewer seriously, we have thoroughly reviewed and revised the manuscript to ensure that the grammar and language used are precise and easy to read.

1. Line 27-28 (Abstract): "To make TimeTalk can be applied to study paracrine" is grammatically incorrect.

Response:

We apologize for the word redundancy. We revised the abstract section according to the reviewer's feedback. The following is the related sentence in the revised version:

"We have incorporated an interpolation strategy into the TimeTalk workflow to ensure its applicability in paracrine settings."

2. Line 58: "The high expression Nanog of progenitor cells in ICM [expression] high levels of Fgf4".

Response:

We apologize for not clearly describing the regulation of PE formation mediated by eLR *Fgf4-Fgfr2* and *Fgf4-Fgfr1*. In addition, we have amended the grammatical error and rewritten the text in the manuscript:

Original: "The high expression *Gata6* of progenitor cells in ICM express high levels of FGF receptor gene *Fgfr1* and *Fgfr2*. The high expression Nanog of progenitor cells in ICM expression high levels of *Fgf4*. The binding of FGF4 secreted by Nanog^{high} cells to FGFR1 or FGFR2 in *Gata6*^{high} cells construct a positive feedback loop with GATA6 to activate PE program gene expression."

Revised: "The progenitor cells in the ICM with high expression of *Gata6* express high levels of the FGF receptor genes *Fgfr1* and *Fgfr2*. Similarly, the progenitor cells in the ICM with high expression of *Nanog* express high levels of *Fgf4*. The binding of FGF4 secreted by Nanog^{high} cells to FGFR1 or FGFR2 in *Gata6*^{high} cells construct a positive feedback loop with GATA6 to activate gene expression of primitive endoderm program."

3. Line 69: "we developed and applied a computational framework named TimeTalk [utilized] temporal series information [though] study the dynamics of autocrine within the embryos"

Response:

We apologize for this oversight. We revised the text in the manuscript:

Original: "To address these issues, we developed and applied a computational framework named TimeTalk utilized temporal series information though study the dynamics of autocrine within the embryos to identify early embryo development-related ligand-receptor pairs (eLRs) from integrated public time-course mouse scRNA-seq datasets."

Revised: "To address these issues, we developed and applied a computational framework named TimeTalk for utilizing temporal series information to study the dynamics of autocrine signaling within the embryos and identify early embryo development-related ligand-receptor pairs (eLRs) from integrated public time-course mouse scRNA-seq datasets."

R1-11. There are more such errors throughout the manuscript. The authors should carefully proofread the manuscript and fix them.

Response:

We are grateful to the reviewer for highlighting the need to improve the language quality. As a result, we have thoroughly proofread the manuscript and made every possible effort to ensure our writing is clear, precise, and accurate.

Reviewer #2

In the present manuscript, Wang et.al. discussed the development "TimeTalk", a tool developed to study cell to cell communication in the embryonic developmental process. Early stages of embryonic development are a complex process of self-assembly where deconvolution of the events leading to this orchestrated process is extremely important. Authors by developing this tool "TimeTalk" have aimed to resolve the events and investigated how cells communicate during the process. Deployment of this computational framework would be in general helpful to probe into the cell-cell communication and the series of events associated with other developmental processes and in disease contexts where this communication and series of events are impaired. The article is well written, and the message is clear. But in many cases contextualization or introduction of some relevant terminologies, are missing; this could a big hindrance for a broader readership. Also, there are segments with grammatical errors, more precisely sentence framing errors. These need to be rectified. Given the strength of the tool developed and the likely impact it would have in deciphering cell-cell communication, I would consider the article for publication upon addressing the issues I highlighted.

Here are few specific major concerns which need to be addressed:

Response:

We greatly appreciate the reviewer for providing positive, insightful, and constructive comments on our study. To make the introduction of related concepts more transparent for the audience, we have revised the manuscript by incorporating relevant terminologies, such as ZGA. Furthermore, we have addressed and incorporated all the comments in the revision.

R2-1 Early embryonic development is a multi-stepped process and the "TimeTalk" framework aims to resolve the events by probing into cell-cell communication. But the authors did not include any proper detailing of the embryonic development, save some very brief outline in the introduction. A little more details would be helpful for the broader readership.

Response:

We appreciate this suggestion and have added more detail about early embryo development in the introduction section, specifically regarding the process of ZGA, as it is related to the validation of eLR. The following paragraph is the background of early embryo development in the introduction section of the revised manuscript.

“Early embryo development is a multi-level regulatory process in which fertilized egg undergoes rounds of cleavage to form a self-organized, hollow sphere structure called blastocyst⁶. At the beginning of the early embryo development, the paternal gametes sperm and maternal gametes oocytes fused to form a one-cell embryo called the zygote in fertilization. Then, the one-cell embryo zygote experiences rounds of cell cleavage division with maternal factor decay and zygotic genome activation (ZGA), which is called maternal to zygotic transition (MZT)⁷. After the fourth cleavage division, the embryo begins to compact into blastomeres. With blastomere formation, there are two critical differentiation events: the first cell fate decision and the second cell fate decision. In the first cell fate decision from the 8-cell stage to the 32-cell stage, the cells in the embryo segregate into two lineages: trophoctoderm (TE) and inner cell mass (ICM), while TE lineage will develop into the placenta. However, in the second cell fate decisions from the early-blastocyst stage to the late-blastocyst stage, the cells in the ICM lineage differentiated into epiblast (EPI) and primitive endoderm (PE) lineage. The EPI lineage will develop into the fetus, and the PE lineage will develop into the Yolk sac^{6,8,9}. The phenomena intrigue evolving research on the mechanisms of the two fate decision events^{9,10}.”

R2-2. The objective sentence in the introduction suddenly brings in the relevance of time-course scRNA-seq datasets; this is hard to comprehend in the introduction section and hence needs some context and a brief introduction.

Response:

We greatly appreciate the reviewers' valuable suggestions. As recommended, we have added the following background information before the objective statement paragraph in the revised version of the manuscript:

"With the advancement of functional genomics research, more cell-cell communication *priori* knowledge resources, including ligand-receptor databases, have been accumulated in recent years¹¹. On the other hand, the development of single-cell RNA sequencing (scRNA-seq) technology has made it possible to infer the cell-cell communication events between different cells. Consequently, integrating single-cell transcriptomic sequencing data and prior knowledge to infer cell-cell communication has emerged as a new research direction in bioinformatics. As for early embryo development studies, research performed scRNA-seq over each development stage to obtain the time-course scRNA-seq data to measure the dynamic changes in cell states and types. However, it is critical to note that definitive cell types do not emerge during early embryo development until blastocyst

formation. However, commonly used tools for cell-cell communication inference, such as CellPhoneDB¹² and CellChat¹³, are designed to study cell-cell communication between given cell types and cannot meet the requirements of early embryo development research. Moreover, the need for multiple time points of cell-cell communication analysis during early embryo development and the need to elucidate potential causal relationships between cell-cell communication and gene regulatory networks during dynamic changes of early embryo development also pose challenges to existing cell-cell communication research."

R2-3. There are sentence framing errors in Introduction (and in other sections of the article).

Response:

We express our gratitude to the reviewer for bringing to our attention the issue raised in the manuscript. We apologize for any confusion caused by the sentence framing errors in the Introduction and other sections of the article. We have carefully reviewed and revised the manuscript to ensure that the sentences are correctly written and clearly convey our intended meaning.

R2-4. Authors while citing the interplay between tTF and e-LR mentioned the LR, and T.F. formed gene regulator network. Authors need to elaborate on this and mention how it is relevant.

Response:

We appreciate the reviewer's suggestions in the first paragraph in the section "The interplay between the tTF and eLR orchestrates early embryo development" from line 241 to line 244 in the original manuscript. Indeed, the LR and T.F. we mentioned formed a gene regulatory network to control the endoderm specification in sea urchin embryos is Wnt8-frizzled and Hox11/13b¹⁴. As in mammals, the interplay between *Fgf4-Fgfr2* and *Gata6* guides the specification of primitive endoderm (PE) lineage. In this article, we are focusing on mammalian embryo development. Thus, we have removed the examples related to the sea urchin embryo and have rewritten the paragraphs in the revised version of the manuscript accordingly:

"As it is previously reported, the interplay between the eLR *Fgf4-Fgfr2* and tTF *Gata6* formed a positive regulatory network to guide the separation of PE and EPI lineage during early embryo development¹⁵. However, no comprehensive probing of the relationship between eLR and tTF has been conceived. Therefore, we hypothesized that investigating the interplay between eLR and TF will help us to understand the gene regulatory network to control mouse embryo development."

R2-5. It would be helpful for the readers if authors justify and elaborate on how cluster 4 and 6 enriched the Hippo signaling pathway.

Response:

We appreciate the keen insight provided by the reviewer. We apologize for not providing a thorough justification of how these clusters enriched the Hippo signaling pathway in the initial version of the manuscript. In the revised manuscript, we elaborate on the enrichment results from 2 aspects:

- Upon analyzing the enrichment results, we discovered that cluster 4 eLR was enriched in the Hippo signaling pathway with 23 eLRs consisting of genes such as *Cdh1*, *Wnt3a*, *Areg*, *Gdf5*, *Wnt5a*, *Wnt7a*, *Fgf1*, *Tgfbr1*, *Fzd2*, *Itgb2*, *Fzd5*, *Fzd1*, *Fzd4*, *Fzd9*, *Bmpr1b*, *Bmpr2*, *Fzd3*, cluster 6 eLR was enriched in the Hippo signaling pathway with the following genes: *Wnt3a*, *Wnt7b*, *Fzd7*, *Fzd5*, *Fzd3*, *Fzd6*. We also listed the related eLR in the following table (Supplementary Fig. 15 a in the revised manuscript, see below). Most of these genes are known to be involved in the TGF- β and Wnt signaling pathways, which have crosstalk with Hippo signaling pathway¹⁶. This crosstalk between Hippo, Wnt, and TGF- β signaling pathway was also indicated in the KEGG database (see the figure below, This figure was obtained from the KEGG database, the web link is <https://www.genome.jp/pathway/mmu04390>)

eLR	L_gene	R_gene	Cluster
Cdh1-Itgb7	Cdh1	Itgb7	cluster_4
Cdh1-Itgae	Cdh1	Itgae	cluster_4
Gdf3-Tgfbr1	Gdf3	Tgfbr1	cluster_4
Tdgf1-Tgfbr1	Tdgf1	Tgfbr1	cluster_4
Wnt3a-Fzd2	Wnt3a	Fzd2	cluster_4
Icam2-Itgb2	Icam2	Itgb2	cluster_4
Tln1-Itgb2	Tln1	Itgb2	cluster_4
Plau-Itgb2	Plau	Itgb2	cluster_4
Areg-Egfr	Areg	Egfr	cluster_4
Gdf5-Ror2	Gdf5	Ror2	cluster_4
Wnt5a-Fzd5	Wnt5a	Fzd5	cluster_4
Wnt7a-Fzd5	Wnt7a	Fzd5	cluster_4
Wnt7a-Fzd1	Wnt7a	Fzd1	cluster_4
Wnt7a-Fzd4	Wnt7a	Fzd4	cluster_4
Wnt7a-Fzd9	Wnt7a	Fzd9	cluster_4
Wnt5a-Fzd4	Wnt5a	Fzd4	cluster_4
Gdf5-Acvr2b	Gdf5	Acvr2b	cluster_4
Gdf5-Bmpr1b	Gdf5	Bmpr1b	cluster_4
Gdf5-Bmpr2	Gdf5	Bmpr2	cluster_4
Fgf1-Fgfr1	Fgf1	Fgfr1	cluster_4
Fgf1-Fgfr2	Fgf1	Fgfr2	cluster_4
Cthrc1-Fzd3	Cthrc1	Fzd3	cluster_4
Cthrc1-Fzd5	Cthrc1	Fzd5	cluster_4
Wnt3a-Fzd7	Wnt3a	Fzd7	cluster_6
Wnt3a-Fzd5	Wnt3a	Fzd5	cluster_6
Wnt3a-Lrp6	Wnt3a	Lrp6	cluster_6
Wnt3a-Fzd3	Wnt3a	Fzd3	cluster_6
Wnt7b-Fzd3	Wnt7b	Fzd3	cluster_6
Wnt3a-Fzd6	Wnt3a	Fzd6	cluster_6

- Next, we attempt to illustrate the potential relationship between the activity of eLR and the activity of the Hippo signaling pathway. According to the literature¹⁷, there are 19 core Hippo signaling pathway genes: *Stk3*, *Taz*, *Lats1*, *Taok2*, *Taok1*, *Lats2*, *Wwc1*, *Tead4*, *Nf2*, *Yap1*, *Stk4*, *Taok3*, *Frmd6*, *Tead2*, *Tead3*, *Sav1*, *Tead1*, *Mob1b*, *Mob1a*. We have identified that the C5 tTF contains the *Tead1* and *Tead4* genes, which are part of the core Hippo pathway gene sets. Therefore, *Tead2* and *Tead4* can be used as indicators of Hippo signaling pathway activity. As for *Tead2*, the eLR *Wnt3a-Fzd5* exhibited an increase in activity over pseudotime, with the Granger test indicating that *Tead2* was responsible for activating *Wnt3a-Fzd5* activity (Supplementary Figure 15b, see below). Prior research has demonstrated that both the *Wnt3a* and *Fzd5* genes are crucial for subsequent trophectoderm lineage development¹⁸. Our findings, combined with this previous research, demonstrate that Hippo signaling pathway activation leads to eLR activation, indicating a potential role for the Hippo signaling pathway in regulating trophectoderm development by activating eLR activity. The eLR *Cthrc1-Fzd3* exhibited a peak expression between the 4-cell stage and 16-cell stages with respect to *Tead4* expression, suggesting that *Tead4* has a feedback relationship with *Cthrc1-Fzd3* activity (Supplementary Figure 15c, see below). Previous studies indicate that *Cthrc1* selectively activates the planar cell polarity pathway of Wnt signaling by stabilizing the Wnt-receptor complex¹⁹. Additionally, *Cthrc1* promotes trophoblast growth, migration, and invasion through reciprocal Wnt/ β -catenin regulation²⁰. Our findings suggest that the cell-cell communication triggered by *Cthrc1* may also interact with the Hippo signaling pathway to promote trophectoderm lineage formation.

The relevant section of the revised version of the manuscript now includes these results.

R2-6. Authors in the discussion should also discuss the scope of the TimeTalk framework in context of other cell-cell communication and in the context of altered states, like diseases.

Response:

We want to express our gratitude to the reviewer for their enlightening suggestions. In response to their feedback, we have included two additional paragraphs in the discussion section of the revised manuscript, which clearly elucidate the scope and applicability of TimeTalk. Specifically, we have emphasized the potential of the TimeTalk framework in unraveling the underlying mechanisms of tumor development and progression. We have also clarified that the current version of TimeTalk is suitable for cell-cell communication that involves ligand-receptor interactions. Furthermore, we have demonstrated that with the advancement of detection technology, the TimeTalk framework can be extended to encompass extracellular vesicle (EV)-mediated cell-cell communication.

R2-7. It would be interesting if authors discuss the ligand receptor interaction in the context of co-evolution of ligand and receptor.

Response:

We appreciate the noteworthy suggestion from the reviewer. The process of co-evolution involves a heritable change in one entity that creates selective pressure for a change in another entity. Such entities can vary from nucleotides and amino acids to proteins, entire organisms, and potentially even ecosystems across the evolutionary time²¹. We agree that the co-evolution of ligand and receptor is an exciting and relevant topic in the context of cell-cell communication.

To tackle these issues, we conducted additional analysis by utilizing the HomoloGene database to obtain homologous ligand and receptor genes in four mammalian species: *Mus Musculus* (mouse), *Rattus norvegicus* (rat), *Bos Taurus* (cattle), and *Homo sapiens* (human). This approach obtained 1119 genes, consisting of 922 ligand-receptor pairs (208 are eLR pairs, 714 are not eLR pairs). Pairwise coding sequence alignment was conducted for each gene in the four species, and the ratio of nonsynonymous and synonymous substitution rates (Ka/Ks) was subsequently calculated. Co-evolution trends were evaluated by computing the Pearson correlation coefficient (PCC) of the Ka/Ks ratio for each ligand-receptor pair in each pairwise species comparison.

Co-evolution analysis was conducted on both eLR and non-eLR pairs, and the results indicate a similar distribution of co-evolution trends between the two groups. (Supplementary Fig. 22 a, as shown below). For instance, the PCC value of Ka/Ks for the eLR pair Fgf10-Fgfr2 was found to be 0.97915, while the non-eLR pair yielded a value of 0.9758 (Supplementary Fig. 22 b-c, as shown below). These results indicate that eLR and non-eLR are exposed to the same selection pressure.

We added these co-evolution analyses in the discussion section of the revised manuscript.

Reviewer #3

In this article, authors proposed a computational frame to track the cell to cell communication through ligand-receptors binding, leads to cell fate during early embryonic development.

The proposed framework is piece of fresh concept addressing an interesting biological questions. The outcomes look promising and relevant enough. Though there are few major concerns

Response:

We are sincerely grateful to the reviewer for their positive feedback regarding the novelty and significance of our proposed framework. We are delighted that the reviewer found our concept valuable in addressing the essential biological questions we aimed to tackle. We appreciate the insightful and constructive comments on our study and have incorporated all of them into the revision.

Major issue:

R3-1. The language of the article is not elucidated enough.

Response:

We have enhanced the clarity and precision of our writing to prevent any ambiguity or misrepresentation. For example, we have rewritten the abstract to improve its grammatical accuracy and readability. The revision of the language was highlighted in the article.

R3-2. Use acronyms carefully.

Response:

We have reduced the acronyms according to the reviewers' suggestions to enhance the readability. For example, we revised the usage of acronyms in the abstract sections:

Original: We found that cell-cell communication between EPI and PE lineage in blastoids share LR pairs and signaling pathways in naturally developed blastocysts.

Revised: Furthermore, by using Timetalk in the blastocyst and blastoid models, we found that the blastoid models share the core communication pathways with the epiblast and primitive endoderm lineages in the blastocysts.

R3-3. Figure legends should be more explanatory

Response:

We appreciate the reviewer's suggestion to improve the explanatory content of our figure legends. In response, we have made a concerted effort to enhance the clarity and

conciseness of our figure legends in the revised version of the manuscript. We aim to ensure that each figure legend provides a clear and informative description of the key findings presented in the corresponding figure. For example, we updated the legend of Figure 4a.:

"Heatmap of the interaction scores of all identified 430 eLRs can be grouped into 6 clusters. In the row annotation of the heatmap, the "cluster" column illustrates the classification of the eLR, C1, C2, C3, C4, C5, and C6 columns illustrate the relationship of each eLR to the C1, C2, C3, C4, C5, C6 regulons. In the column annotation of the heatmap, the "Stage" row illustrates the development stage of every single cell, the "Pseudotime" row illustrates the pseudotime, C1, C2, C3, C4, C5, C6 row illustrates the activity of every single cell."

R3-4. The flow of the summary is little disconnected

Response:

We are grateful for the suggestions about the flow of the summary. We have revised it with improved coherence and connectivity by reorganizing sections and adding transition phrases. We hope these changes have addressed the reviewer's concerns and improved the manuscript's clarity and readability.

Reference

1. Ramskold, D. *et al.* Full-length mRNA-Seq from single-cell levels of RNA and individual circulating tumor cells. *Nat Biotechnol* **30**, 777-82 (2012).
2. Deng, Q., Ramskold, D., Reinius, B. & Sandberg, R. Single-cell RNA-seq reveals dynamic, random monoallelic gene expression in mammalian cells. *Science* **343**, 193-6 (2014).
3. Büttner, M., Miao, Z., Wolf, F.A., Teichmann, S.A. & Theis, F.J. A test metric for assessing single-cell RNA-seq batch correction. *Nat Methods* **16**, 43-49 (2019).
4. Zhang, B. *et al.* Allelic reprogramming of the histone modification H3K4me3 in early mammalian development. *Nature* **537**, 553-557 (2016).
5. Schrode, N., Saiz, N., Di Talia, S. & Hadjantonakis, A.K. GATA6 levels modulate primitive endoderm cell fate choice and timing in the mouse blastocyst. *Dev Cell* **29**, 454-67 (2014).
6. Zhu, M. & Zernicka-Goetz, M. Principles of Self-Organization of the Mammalian Embryo. *Cell* **183**, 1467-1478 (2020).
7. Schulz, K.N. & Harrison, M.M. Mechanisms regulating zygotic genome activation. *Nat Rev Genet* **20**, 221-234 (2019).
8. Wang, H. & Dey, S.K. Roadmap to embryo implantation: clues from mouse models. *Nat Rev Genet* **7**, 185-99 (2006).
9. Rossant, J. & Tam, P.P. Blastocyst lineage formation, early embryonic asymmetries and axis patterning in the mouse. *Development* **136**, 701-13 (2009).
10. Zernicka-Goetz, M., Morris, S.A. & Bruce, A.W. Making a firm decision:

- multifaceted regulation of cell fate in the early mouse embryo. *Nat Rev Genet* **10**, 467–77 (2009).
11. Armingol, E., Officer, A., Harismendy, O. & Lewis, N.E. Deciphering cell-cell interactions and communication from gene expression. *Nat Rev Genet* **22**, 71–88 (2021).
 12. Vento-Tormo, R. *et al.* Single-cell reconstruction of the early maternal-fetal interface in humans. *Nature* **563**, 347–353 (2018).
 13. Jin, S. *et al.* Inference and analysis of cell-cell communication using CellChat. *Nat Commun* **12**, 1088 (2021).
 14. Peter, I.S. & Davidson, E.H. A gene regulatory network controlling the embryonic specification of endoderm. *Nature* **474**, 635–9 (2011).
 15. Chazaud, C. & Yamanaka, Y. Lineage specification in the mouse preimplantation embryo. *Development* **143**, 1063–74 (2016).
 16. Attisano, L. & Wrana, J.L. Signal integration in TGF- β , WNT, and Hippo pathways. *F1000Prime Rep* **5**, 17 (2013).
 17. Wang, Y. *et al.* Comprehensive Molecular Characterization of the Hippo Signaling Pathway in Cancer. *Cell Rep* **25**, 1304–1317.e5 (2018).
 18. Dietrich, B., Haider, S., Meinhardt, G., Pollheimer, J. & Knöfler, M. WNT and NOTCH signaling in human trophoblast development and differentiation. *Cell Mol Life Sci* **79**, 292 (2022).
 19. Yamamoto, S. *et al.* Cthrc1 selectively activates the planar cell polarity pathway of Wnt signaling by stabilizing the Wnt-receptor complex. *Dev Cell* **15**, 23–36 (2008).
 20. Li, Y. *et al.* CTHRC1 promotes growth, migration and invasion of trophoblasts via reciprocal Wnt/ β -catenin regulation. *J Cell Commun Signal* **16**, 63–74 (2022).
 21. Fraser, H.B., Hirsh, A.E., Wall, D.P. & Eisen, M.B. Coevolution of gene expression among interacting proteins. *Proc Natl Acad Sci U S A* **101**, 9033–8 (2004).

REVIEWERS' COMMENTS:

Reviewer #1 (Remarks to the Author):

The authors have satisfactorily answered my previous questions. There are some remaining grammar/typo issues that need to be fixed. For example:

(1) Line 311-313: "it was found that the C5 tTF contained Tead2 and Tead4, ..., were present in the C5 tTF" - incorrect grammar.

(2) Line 378, remove the word "While".

(3) Line 567, "Supplementary Fig. xxxa": what is xxx?

(4) Line 568, "Supplementary Fig. xxxb": what is xxx?

Reviewer #2 (Remarks to the Author):

Authors have carefully gone through all the recommendations. They have addressed most of the points, rephrased many of the sections in the article and added new relevant data as per all the reviewer's recommendations. The rephrased sections provide clarity to the inferences drawn and the new discussion is substantially improved. With all the modifications I think the article would be an interesting read for the community. I recommend the article for acceptance.

REVIEWERS' COMMENTS:

Reviewer #1 (Remarks to the Author):

The authors have satisfactorily answered my previous questions.

There are some remaining grammar/typo issues that need to be fixed. For example:

(1) Line 311-313: "it was found that the C5 tTF contained Tead2 and Tead4, ..., were present in the C5 tTF" - incorrect grammar.

(2) Line 378, remove the word "While".

(3) Line 567, "Supplementary Fig. xxxa": what is xxx?

(4) Line 568, "Supplementary Fig. xxxb": what is xxx?

Reviewer #2 (Remarks to the Author):

Authors have carefully gone through all the recommendations. They have addressed most of the points, rephrased many of the sections in the article and added new relevant data as per all the reviewer's recommendations. The rephrased sections provide clarity to the inferences drawn and the new discussion is substantially improved. With all the modifications I think the article would be an interesting read for the community. I recommend the article for acceptance.

Point-to-point response

Reviewer #1 (Remarks to the Author):

The authors have satisfactorily answered my previous questions. There are some remaining grammar/typo issues that need to be fixed. For example:

Response:

We sincerely express our great gratitude to Reviewer #1 for helping us improve the quality of this manuscript. Your dedication and commitment to reviewing our work are truly appreciated. We are encouraged to learn that we have satisfactorily addressed your previous concerns. We have now fixed the following problems and carefully proofread the manuscript.

(1) Line 311-313: "it was found that the C5 tTF contained *Tead2* and *Tead4*, ..., were present in the C5 tTF" - incorrect grammar.

Response:

Thank you for highlighting this error. We have split the sentence and removed the unnecessary phrase "were present in the C5 tTF". We have revised the sentence as follows:

Original sentence: "The potential relationship between eLR activity and the Hippo signaling pathway was also explored, and it was found that the C5 tTF contained *Tead2* and *Tead4*, which are part of the core Hippo pathway gene sets were present in the C5 tTF."

Revised sentence: "An investigation was conducted to determine whether there is a connection between eLR activity and the Hippo signaling pathway. It was discovered that the C5 tTF includes *Tead2* and *Tead4*, which are part of the core Hippo pathway gene sets."

(2) Line 378, remove the word "While".

Response:

We have now modified the sentence following the advice raised by Reviewer #1.

Original sentence: "While TimeTalk is quite sensitive to the parameter *winsz* (Supplementary Fig. 19a), but relatively stable to *numPts* and *lags* parameters (Supplementary Fig. 19b-c)."

Revised sentence: "TimeTalk is quite sensitive to the parameter *winsz* (Supplementary Fig. 19a), but relatively stable to *numPts* and *lags* parameters (Supplementary Fig. 19b-c)."

(3) Line 567, "Supplementary Fig. xxxa": what is xxx?

Response:

We sincerely apologize for the mistake regarding the temporary notation of the supplementary figure number during the organization of our previously revised manuscript. We regret that we overlooked updating this notation in our previous submission. Here, "Supplementary Fig. xxxa" refers to Supplementary Fig. 7a. We have now fixed this error.

Original sentence: "Thus, we select the vertical coordinate of point D as the cutoff value, which is 0.0987, for positive co-vary LR (Supplementary Fig. xxxa)."

Revised sentence: "Thus, we select the vertical coordinate of point D as the cutoff value, which is 0.0987, for positive co-vary LR (Supplementary Fig. 7a)."

(4) Line 568, "Supplementary Fig. xxx b": what is xxx?

Response:

We apologize for this mistake. As explained in previous issues, here we have fixed this error.

Original sentence: "Using the same process, the cutoff for the negative co-vary LR is - 0.0587 (Supplementary Fig. xxx b)."

Revised sentence: "Using the same process, the cutoff for the negative co-vary LR is - 0.0587 (Supplementary Fig. 7b)."

Furthermore, we have carefully proofread the manuscript again to prevent any mistakes.

Reviewer #2 (Remarks to the Author):

Authors have carefully gone through all the recommendations. They have addressed most of the points, rephrased many of the sections in the article and added new relevant data as per all the reviewer's recommendations. The rephrased sections provide clarity to the inferences drawn and the new discussion is substantially improved. With all the modifications I think the article would be an interesting read for the community. I recommend the article for acceptance.

Response:

We would like to express our sincere appreciation to Reviewer #2 for their positive and supportive feedback on our revised manuscript. We are thrilled to hear the credit from Reviewer #2 for our diligent efforts in addressing all the reviewer's suggestions and integrating new information into the manuscript. We are grateful for your recommendation

to accept our article for publication and your comment on its potential appeal to the community. It is truly an honor to receive such encouraging feedback from a respected reviewer. In this revision, we have proofread the manuscript to ensure its overall quality.